# FLASHBACK: MEMORY-DRIVEN ZERO-SHOT, REAL-TIME VIDEO ANOMALY DETECTION

## ABSTRACT

Video anomaly detection (VAD) aims to identify unusual events in continuous video streams, yet most existing systems either rely on domain-specific retraining or fail to meet strict real-time demands. We present **Flashback**, a zero-shot and real-time paradigm that reframes VAD as retrieval over an offline pseudo-scene memory. Inspired by how humans recall past experiences to judge the present, Flashback constructs a large set of normal and anomalous captions entirely offline with a language model, embeds them once with a frozen video-text encoder, and reuses this memory online. At inference, each segment is matched against the memory to produce both an anomaly score and a textual rationale, eliminating all online LLM calls and sustaining per-segment deadlines. Three lightweight controls improve robustness: *repulsive prompting* separates normal and anomalous caption spaces, *scaled anomaly penalization* corrects residual anomaly bias, and *certainty-driven runtime encoder selection* maintains weakly-hard real-time guarantees by allocating extra compute only to difficult segments. On UCF-Crime and XD-Violence, Flashback achieves 87.7 AUC and 75.0 AP, outperforming prior zero-shot methods while providing human-readable explanations at up to 43.8 fps on a single consumer GPU. The result is the first VAD system that is simultaneously zero-shot, real-time, and explainable.

## 1 INTRODUCTION

Video anomaly detection (VAD) aims to automatically identify events that deviate from learned normal patterns in continuous video streams, alleviating the impracticality of manual monitoring in public safety (Zhu et al., 2021), intelligent transportation (Bogdoll et al., 2022), and industrial inspection systems (Roth et al., 2022). The worldwide proliferation of surveillance cameras is generating video volumes that far exceed human monitoring capacity, making timely detection at human scale impractical (BusinessWire, 2023; Grand View Research, 2024; Sultani et al., 2018). Automated VAD has therefore become a practical necessity.

Real-world deployment is hindered by two fundamental obstacles: *domain dependency* and *real-time constraints*. First, most VAD paradigms, whether weakly-supervised (Sultani et al., 2018; Ye et al., 2024), one-class (Hasan et al., 2016; Lu et al., 2013), or unsupervised (Thakare et al., 2023a; Tur et al., 2023a), require collecting and annotating domain-specific footage followed by model retraining for each new environment, which is prohibitive when target-domain samples are scarce or unavailable. Second, because emergencies can arise at any time, inference for each fixed-length segment must complete within its duration to prevent latency accumulation; otherwise, continuous streams force the system to skip segments, undermining uninterrupted VAD.

Recent work has addressed domain generalization and real-time processing separately. Zero-shot VAD leverages pre-trained vision-language models (VLMs) (Liu et al., 2023) and large language models (LLMs) (Touvron et al., 2023) to avoid domain-specific retraining. Caption-and-score approaches (Gong & et al., 2024; Zanella et al., 2024) generate frame-level captions with a VLM and then compute anomaly scores with an LLM, but incur substantial autoregressive captioning overhead and are susceptible to noisy text outputs. Prompt-based methods (Ahn et al., 2025; Wu & et al., 2024; Ye et al., 2024) reduce LLM calls by injecting learned prompts into VLM inference, improving efficiency and transferability; however, many retain inference-time LLM dependence, struggle to enforce temporally coherent scoring across contiguous frames, and remain sensitive to prompt

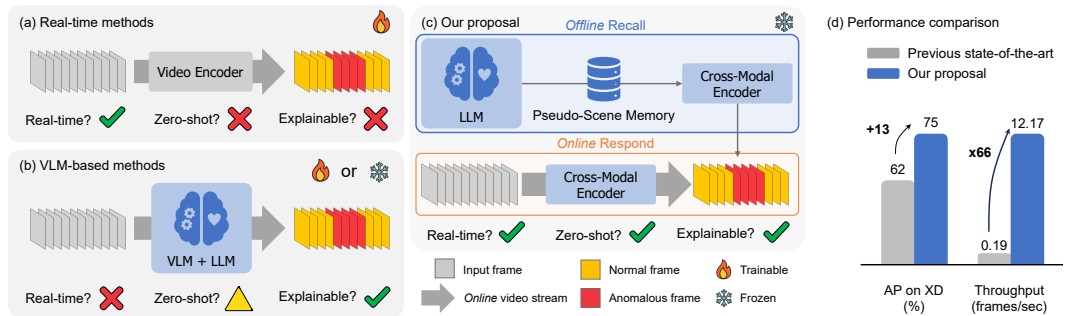

Figure 1: **Bridging speed and reasoning.** (a) Real-time VAD keeps a light video encoder online but cannot work zero-shot or explain its decisions. (b) VLM-based VAD adds large VLM and LLM in the loop; reasoning is possible, yet speed drops and zero-shot ability is partial. (c) Flashback moves the LLM offline, builds a pseudo-scene memory once, and uses a frozen cross-modal encoder at test time, so it is simultaneously real-time, zero-shot, and explainable. (d) On XD-Violence (Wu et al., 2020) (XD), this design lifts AP by 13 and boosts throughput 66 times over the prior state-of-the-art.

design. In parallel, real-time VAD aims to process each segment before the next arrives. End-to-end weakly-supervised models (Karim et al., 2024) accelerate inference but still fall short on unseen domains, while density-estimation detectors (Micorek et al., 2024) achieve per-segment delays around $200\,\mathrm{ms}$ yet require domain-specific updates to maintain accuracy. Existing approaches treat these axes separately; none satisfies *all* of the following under a single, unified design: (i) target-domain-free training (zero-shot), (ii) strict streaming latency (latency $< T$), and (iii) no inference-time LLM calls.

In this paper, we introduce Flashback, a paradigm that unifies cross-domain generalization with consistently low-latency inference. As illustrated in Figure 1, Flashback comprises two stages: (i) *offline memory construction* and (ii) *online caption-retrieval inference*. Offline, a frozen LLM (OpenAI, 2024) synthesizes a broad spectrum of normal and anomalous captions without any video input; each caption is then embedded by a frozen cross-modal encoder (Bolya et al., 2025), and caption-embedding pairs are stored in a text-only memory. Online, the incoming stream is partitioned into fixed-length segments of length $T$; segment embeddings are matched against the memory via similarity search to produce segment-level anomaly scores, which are temporally smoothed into frame-level predictions.

Despite the streamlined pipeline, three challenges arise. (i) *Anomalous-prior shift*: the cross-modal encoder semantically biases normal content toward anomalous captions. We introduce *Repulsive Prompting (RP)*, using distinct templates for normal vs. anomalous captions to widen inter-class embedding margins. (ii) *Residual anomaly skew at inference*: even with RP, similarity scores may remain skewed toward anomalies. We apply *Scaled Anomaly Penalization (SAP)*, rescaling anomalous caption embedding magnitudes prior to similarity search to stabilize the score distribution. (iii) *Balancing accuracy and speed*: scene difficulty varies, creating a latency-accuracy trade-off between a fast small encoder and a slower but more accurate large encoder. We develop a *certainty-driven runtime encoder selection* that estimates segment-level certainty from retrieval statistics and switches encoders online with hysteresis and queue-slack checks, preserving the strict streaming constraint.

By removing all LLM calls from the online loop, Flashback deploys without any additional data collection or fine-tuning. Its retrieval-based pipeline explains decisions with human-readable captions and keeps pace with the stream by completing each fixed-length segment before the next begins even when segments are as short as one second. On UCF-Crime (Sultani et al., 2018) and XD-Violence (Wu et al., 2020), Flashback outperforms strong zero-shot, unsupervised, and one-class baselines while meeting real-time requirements.

The contributions of this work are as follows:

**Unified zero-shot and real-time VAD.** We recast video anomaly detection as video-to-text retrieval over a text-only pseudo-scene memory synthesized offline without videos, eliminating inference-time language model calls and enabling deployment without additional data collection or fine-tuning.

**Techniques to mitigate anomaly bias.** We introduce a prompting scheme that deliberately separates normal and anomalous caption spaces to reduce encoder-side bias, and a score-calibration strategy that attenuates anomalous-caption dominance during retrieval.

**Streaming-safe runtime encoder selection.** We design a certainty-driven runtime model selection mechanism that switches between a fast small encoder and an accurate large encoder using lightweight statistics and hysteresis/queue checks, maintaining per-segment deadlines in continuous streams.

**Interpretable, real-time results on public benchmarks.** The retrieval pipeline produces human-readable explanations via the matched captions while meeting real-time requirements, and it out-performs strong zero-shot, unsupervised, and one-class baselines on UCF-Crime and XD-Violence.

## 2 RELATED WORK

**Video anomaly detection.** Early video anomaly detection (VAD) methods minimize reconstruction or other generative losses on normal-only or unlabeled footage (Hasan et al., 2016; Lu et al., 2013; Thakare et al., 2023a; Tur et al., 2023a;b; Wang & Cherian, 2019; Zaheer et al., 2022) and thus treat large reconstruction errors as a signal of abnormality. A parallel line formulates VAD as weakly-supervised multiple-instance learning, using video-level labels (Sultani et al., 2018; Zhang et al., 2019; Wu & Liu, 2021). To broaden coverage, later work incorporates audio cues (Wu et al., 2020; 2025) or instruction-tunes detectors on a privately collected video-caption corpus (Zhang et al., 2024). More recently, researchers exploit the semantic priors of LLMs or VLMs: prompt tuning (Ye et al., 2024) or lightweight adapters (Zhang et al., 2024) improve accuracy and even yield textual rationales. However, all of these approaches still need target-domain videos or captions for fine-tuning; collecting and training on that data consume both time and significant computational costs. In contrast, Flashback requires no additional data or gradient updates yet matches the accuracy of tuned systems.

**Vision-language models for zero-shot VAD.** Large language models (Brown et al., 2020; Ouyang et al., 2022; Wang et al., 2023; Wei et al., 2022a;b) and vision-language models (Alayrac et al., 2022; Dai et al., 2023; Li et al., 2023b; Peng et al., 2024) already achieve strong few- or zero-shot accuracy on language benchmarks (Clark & Etzioni, 2018; Joshi et al., 2017; Kwiatkowski & Palmer, 2019; Paperno et al., 2016; Talmor et al., 2019) and multimodal tasks such as visual question answering (VQA) (Antol et al., 2015; Goyal et al., 2017; Hudson & Manning, 2019; Marino et al., 2019). LAVAD (Zanella et al., 2024) extends this power to anomaly detection: it captions every video segment with a VLM and scores each caption with an LLM, achieving competitive zero-shot performance and human-readable explanations. Subsequent work refines prompts (Ye et al., 2024), adds graph modules for modeling temporal structure (Shao et al., 2025), or fuses audio via audio-language models (Dev et al., 2024). Nevertheless, all of these pipelines keep an auto-regressive model within the repetitive operation, so inference slows to roughly below one frame per second, and latency varies with output caption length. On the contrary, Flashback leverages the knowledge base of an LLM (OpenAI, 2024) *offline*: we generate a pseudo-scene memory without any visual input, store it once, and then perform real-time retrieval with a video-text encoder. At inference time, we only look up the most similar caption for each segment, assign its anomaly label, and use the caption itself as an immediate textual explanation, sustaining 43.82 fps on a single commercial GPU while avoiding any online LLM calls.

## 3 THE FLASHBACK PARADIGM

Drawing on how humans instantly detect and explain anomalies by recalling past experiences (Bar, 2007; Friston, 2005; Summerfield & de Lange, 2014), Flashback replaces *train-then-detect* with *recall-and-retrieve* at a high level: instead of learning from target footage, it detects anomalies by comparing the present to a broad prior of scenes prepared offline. The system has three layers. *Offline pseudo-scene memory* builds a text-only library: a language model synthesizes concise captions for normal and anomalous situations, and a frozen cross-modal encoder embeds them once into a deployment-ready memory. *Online retrieval & scoring* encodes each segment, retrieves nearest captions via similarity search, and uses the matches to produce (i) an anomaly score by aggregating labels and (ii) human-readable rationales. Two lightweight controls improve reliability: class-specific templates keep normal and anomalous representations apart, and a small pre-retrieval down-weighting prevents anomalous captions from dominating the score. Finally, a *deadline-compliant runtime encoder selection* uses certainty with hysteresis and queue-slack checks to engage a large encoder only on hard segments. The result is a zero-shot, real-time, interpretable VAD that avoids LLM calls at inference and keeps online cost to one encoder pass plus a few vector operations.

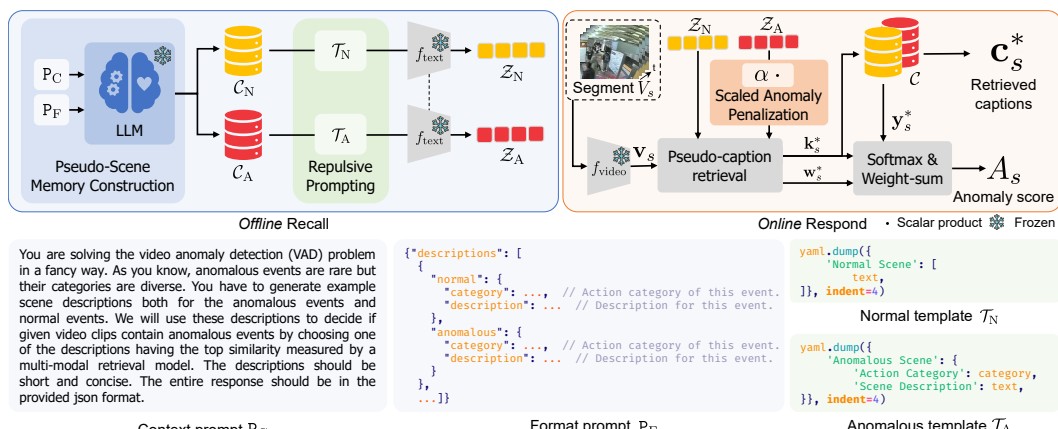

Figure 2: **Overview of Flashback.** Flashback reframes video anomaly detection as a video-to-text retrieval task. **Offline Recall:** concise captions covering normal/anomalous situations are generated once and embedded to form a text-only pseudo-scene memory $\mathcal{C}$. **Online Respond:** each incoming segment $V_s$ is encoded, nearest captions are retrieved, their labels are aggregated into a segment score, and matched texts serve as rationales; causal smoothing yields frame-wise alerts. Inference is zero-shot, LLM-free, and designed to meet per-segment deadlines.

## 3.1 PROBLEM STATEMENT: ZERO-SHOT, REAL-TIME VAD

Let a video stream be $V = \{I_t\}_{t \geq 1}$. The task of video anomaly detection (VAD) is to assign an anomaly score $p_t \in [0, 1]$ to every frame $I_t$. We consider VAD in *camera-network streams*, where frames arrive continuously and decisions must be made online without relying on target-domain data. Let the stream be sampled at $r$ fps. In practice, operators act on short fixed segments that provide the minimal temporal context a single frame lacks, so we process the stream in non-overlapping *segments* of duration $T_{\text{segment}}$ seconds; the $s$-th segment is

$$V_s = \{ I_t \mid s \cdot T_{\text{segment}} \leq t/r < (s+1) \cdot T_{\text{segment}} \}.$$

A detector receives no videos or labels from the deployment domain (*zero-shot*) and must, at run time, assign frame-wise anomaly scores while producing a brief, human-friendly rationale per segment.

**Weakly-hard real-time constraints.** Hard real-time constraints demand that every deadline is met. This is often infeasible in practice because workloads vary and resources may be temporarily overloaded. Weakly-hard real-time (WHRT) (Bernat et al., 2001) constraints relax this requirement by allowing a limited number of deadline misses, while still giving predictable guarantees. A system satisfies a WHRT constraint if it tolerates occasional deadline misses under a bounded pattern. Formally, let $\tau_s$ be the wall-clock time to compute the segment output $(A_s, \mathbf{c}_s^*)$ (an anomaly score and textual explanation) using information up to the end of $V_s$. Given a window of $\xi$ consecutive segments, at most $x$ segments may violate the deadline

$$\tau_s \leq T_{\text{segment}} \quad \text{for all but at most } x \text{ segments in any window of } \xi.$$

This ensures that backlog is bounded and no long-term skipping occurs, while still permitting controlled violations. Such guarantees are practical in deployments where occasional latency is acceptable as long as the majority of actionable events are detected on time.

**Strictly domain-agnostic deployment.** During development, no videos, labels, or metadata from the target domain are used; at run time, model parameters and the offline pseudo-scene memory remain fixed (no fine-tuning, calibration, or test-time adaptation). Our goal is to produce frame-wise scores and textual per-segment rationales that simultaneously (i) respect this domain-agnostic setting and (ii) meet the per-segment deadline, while achieving high detection quality on unseen streams.

## 3.2 OFFLINE RECALL: MEMORY CONSTRUCTION

Zero-shot deployment precludes any site-specific data or calibration yet operators still expect decisions they can read and trust. We therefore construct entirely offline a text-only pseudo-scene memory made

of concise captions embedded by a frozen text encoder. So, inference requires no LLM calls and remains domain agnostic.

**Pseudo-scene memory.** This step builds the offline memory queried at inference time. We run an LLM with two prompts. *Context prompt* $P_C$: The model is told to act as a VAD assistant. It must produce short captions for both normal and anomalous events. $P_C$ also explains that the captions will later be ranked by a cross-modal encoder, so they should be concise and informative. *Format prompt* $P_F$: To keep the output machine-parsable, we supply a schema with two fields, `"normal"` and `"anomalous"`, each holding an event `"category"` and a free-form `"description"`.

The LLM returns ordered lists of captions $\mathcal{C}_N = (c_1^N, \ldots, c_M^N)$ (normal) and $\mathcal{C}_A = (c_1^A, \ldots, c_L^A)$ (anomalous), together with their category lists $\mathcal{K}_N = (\kappa_1^N, \ldots, \kappa_M^N)$ and $\mathcal{K}_A = (\kappa_1^A, \ldots, \kappa_L^A)$. We concatenate them to form $\mathcal{C} = \mathcal{C}_N \oplus \mathcal{C}_A, \mathcal{K} = \mathcal{K}_N \oplus \mathcal{K}_A, \mathcal{Y} = (0, \ldots, 0, 1, \ldots, 1)$, where $\mathcal{Y}$ stores the binary anomaly flags. All captions are generated without any video and encoded once with the text branch $f_{\text{text}}$ and cached, so no further LLM calls or fine-tuning are required during inference.

**Repulsive prompting.** Pseudo-captions for similar scenes often lie close in embedding space even when one is normal and the other anomalous. We introduce repulsive prompting (RP) to separate the two groups. Each caption inserts a keyword: `"Normal"` for normal events and `"Anomalous"` for abnormal ones, then passes through a template. The combined keyword-plus-wrapper defines $\mathcal{T}_N$ and $\mathcal{T}_A$ with $\mathcal{T}_N \neq \mathcal{T}_A$. Encoding these templated captions yields $\mathcal{Z}_N = f_{\text{text}}(\mathcal{T}_N(\mathcal{C}_N, \mathcal{K}_N))$ and $\mathcal{Z}_A = f_{\text{text}}(\mathcal{T}_A(\mathcal{C}_A, \mathcal{K}_A))$. While conceptually related to representation separation in metric and contrastive learning (Hadsell et al., 2006; Schroff et al., 2015), RP differs by using anomaly flags from pseudo-scene memory and requiring no training. It thus offers a lightweight, supervised prompt-based mechanism that enforces separation in the caption space with minimal overhead.

### 3.3 Online respond: retrieval & scoring (inference)

Meeting a strict streaming deadline calls for reducing online work to a single video-encoder pass followed by lightweight similarity search. We embed the current window, retrieve the nearest captions, and aggregate their labels into a score and rationale, while a small pre-retrieval attenuation of anomalous captions stabilizes thresholds under shift.

**Scaled anomaly penalization.** We observe that caption vectors for anomalous events tend to form smaller angles with video features than normal captions do. Therefore, shrinking those vectors before dot product computation is more effective than clipping the score after retrieval. To dampen the inherent bias towards anomalous captions, we rescale their embeddings before retrieval: for every $\mathbf{t}_j \in \mathcal{Z}_A$, we set $\tilde{\mathbf{t}}_j = \alpha \mathbf{t}_j$ with $\alpha \in (0, 1)$. The factor $\alpha$ lowers the magnitude of dot products for anomalous captions, reducing spurious matches with little added computational overhead.

**Pseudo-caption retrieval and scoring.** We chop a test video into non-overlapping segments $\{V_s\}$ of $T_{\text{segment}}$ seconds. We obtain video features of $s$-th segment $\mathbf{v}_s = f_{\text{video}}(V_s)$. For every caption embedding $\mathbf{t}_j \in \mathcal{Z}$, we compute the dot products $\sigma_{s,j} = \mathbf{v}_s^\top \mathbf{t}_j$. We keep the indices of the $K$ largest products $k \in \mathbf{k}_s^* = \arg \text{topK}(\boldsymbol{\sigma}_s; K)$ and convert their products into weights $\mathbf{w}_s^*$ by applying softmax over $k$. Then, we obtain the segment-level anomaly score $A_s$ as the weighted average of retrieved anomaly flags $\mathbf{y}_s^*$. The retrieved captions $\mathbf{c}_s^*$ are returned as instant textual explanation of the segment.

**Frame-level score refinement.** Let $\mathcal{S}_t$ be the index set of segments that cover frame $t$. We average their scores $p_t = \frac{1}{|\mathcal{S}_t|} \sum_{s \in \mathcal{S}_t} A_s$ and convolve the resulting sequence with a 1-D Gaussian kernel. The smoothed curve $\{\tilde{p}_t\}_{t=1}^T$ is our final frame-wise prediction.

### 3.4 Certainty-driven runtime encoder selection

We route only the *hard* segments to the *large* encoder and keep the rest on the *small*; here, "hard" means low certainty and sufficient time budget to meet the deadline. For segment $V_s$, we compute a certainty score $\lambda_s$ from retrieval statistics using the log-likelihood of a two-state Kalman filter (Kalman, 1960), viewed as a recursive Bayesian model (Masreliez & Martin, 1977). The model uses transition and observation operators $\mathbf{F} = \begin{bmatrix} 1 & \Delta t \\ 0 & \phi \end{bmatrix}$, $\mathbf{H} = \begin{bmatrix} 1 & 0 \end{bmatrix}$, with $\phi = e^{-\Delta t / \tau}$, process noise $\mathbf{Q}$, and observation noise $R$. The log-likelihood $\lambda_s = -\frac{1}{2}\left(\tilde{a}_s^2/S_s + \log S_s + \log 2\pi\right)$ quantifies conformity of the score to expected dynamics (derivation in Section D). Adaptive thresholds from history length $H$ are

$$\lambda_s^{\text{low}} = Q_{q_{\text{bottom}}}\big(\{\lambda_u\}_{u=s-H}^{s-1}\big), \quad \lambda_s^{\text{high}} = Q_{q_{\text{top}}}\big(\{\lambda_u\}_{u=s-H}^{s-1}\big),$$

Table 1: **Comparison with state-of-the-art video anomaly detectors on UCF-Crime (Sultani et al., 2018) and XD-Violence (Wu et al., 2020).** Methods are grouped by supervision level. Flashback attains the highest accuracy on both datasets and is the first approach that is simultaneously zero-shot, real-time, and explainable. **Bold** numbers mark the top result. Scores marked [*] are reported by CLIP-TSA (Joo et al., 2023), and scores marked [†] are reported by VERA (Ye et al., 2024).

| Method | Explainable? | Real-time? | UCF-Crime AUC (%) | XD-Violence AP (%) | XD-Violence AUC (%) |
|---|---|---|---|---|---|
| *Weakly-Supervised* | | | | | |
| Sultani *et al.* (Sultani et al., 2018) | | ✓ | 77.92 | - | - |
| GCL (Zaheer et al., 2022) | | ✓ | 79.84 | - | - |
| Wu *et al.* (Wu et al., 2020) | | ✓ | 82.44 | 73.20 | - |
| RTFM (Tian et al., 2021) | | ✓ | 84.03 | 77.81 | - |
| Wu & Liu (Wu & Liu, 2021) | | ✓ | 84.89 | 75.90 | - |
| MSL (Li et al., 2022) | | ✓ | 85.62 | 78.58 | - |
| S3R (Wu et al., 2022) | | ✓ | 85.99 | 80.26 | - |
| MGFN (Chen et al., 2023) | | ✓ | 86.98 | 80.11 | - |
| CLIP-TSA (Joo et al., 2023) | | ✓ | 87.58 | 82.17[*] | - |
| VadCLIP (Wu et al., 2024) | | ✓ | 88.02 | 84.51 | - |
| Holmes-VAD (Zhang et al., 2024) | ✓ | | 84.61[†] | 84.96[†] | - |
| VERA (Ye et al., 2024) | ✓ | | 86.55 | 70.54 | 88.26 |
| *One-Class* | | | | | |
| Hasan *et al.* (Hasan et al., 2016) | | ✓ | - | - | 50.32 |
| Lu *et al.* (Lu et al., 2013) | | ✓ | - | - | 53.56 |
| BODS (Wang & Cherian, 2019) | | ✓ | 68.26 | - | 57.32 |
| GODS (Wang & Cherian, 2019) | | ✓ | 70.46 | - | 61.56 |
| *Unsupervised* | | | | | |
| GCL (Zaheer et al., 2022) | | ✓ | 74.20 | - | - |
| Tur *et al.* (Tur et al., 2023b) | | ✓ | 65.22 | - | - |
| Tur *et al.* (Tur et al., 2023a) | | ✓ | 66.85 | - | - |
| DyAnNet (Thakare et al., 2023b) | | ✓ | 79.76 | - | - |
| RareAnom (Thakare et al., 2023a) | | ✓ | - | - | 68.33 |
| *Zero-shot* | | | | | |
| LAVAD (Zanella et al., 2024) | ✓ | | 80.28 | 62.01 | 85.36 |
| **Flashback**$_B$ **(Ours)** | ✓ | ✓ | 82.05 | 69.40 | 88.13 |
| **Flashback**$_L$ **(Ours)** | ✓ | ✓ | **87.74** | **75.00** | **90.51** |

with $0 < q_{\text{bottom}} < q_{\text{top}} < 1$ and $Q_p(\cdot)$ the empirical quantile. We route $V_s$ to *large* if $\lambda_s \leq \lambda_s^{\text{low}}$ and time budget allows, revert to *small* if $\lambda_s \geq \lambda_s^{\text{high}}$ or budget is tight, and otherwise keep the previous choice. These conditions upgrade only difficult segments when feasible, revert on easy ones, and add hysteresis to avoid flip-flop. Quantile-based normalization makes the switch domain-agnostic, while the budget gate ensures deadlines are met.

## 4 EXPERIMENTAL RESULTS

We structure our evaluation to validate all core properties of Flashback: zero-shot SOTA accuracy without any target-domain training (Section 4.2), real-time property (Section 4.3), and explainability via retrieved captions (Section 4.4 & Section 4.6). We also examine anomaly-score correlation with true anomalousness (Section 4.6), and assess how repulsive prompting (RP) and scaled anomaly penalization (SAP) reduce false positives, as well as SAP's sensitivity to the scale factor $\alpha$ (Section 4.5).

### 4.1 EXPERIMENTAL SETUP

**Datasets.** We evaluate mainly on two large-scale benchmarks. **UCF-Crime** (Sultani et al., 2018) (UCF) contains 290 test videos (140 abnormal). **XD-Violence** (Wu et al., 2020) (XD) provides 800 test videos (500 abnormal). To further demonstrate the expandability of our method, we also evaluate on two datasets that are smaller but define relatively naive anomalies: **ShanghaiTech** (Liu et al., 2018) with 107 test videos and **StreetScene** (Ramachandra & Jones, 2020) with 35 test videos, both of which contain anomalous events in every video.

**Accuracy metrics.** Following prior work (Ye et al., 2024; Zanella et al., 2024) we report the area under the frame-level ROC curve (AUC) for all the datasets and the average precision (AP) for XD-Violence.

**Performance metrics.** We measure median and tail latencies ($p50, p95, p99$), which represent the 50th, 95th, and 99th percentiles of per-segment response times (Dean & Barroso, 2013). Deadline satisfaction follows the weakly-hard real-time model $x^*/\xi$, where at most $x$ out of any $\xi$ consecutive tasks may miss their deadlines (Bernat et al., 2001). We further report the real-time factor (RTF), defined as the ratio of processing time to input duration, with RTF $< 1$ indicating faster-than-real-time operation (Gaido et al., 2020), and throughput, defined as the number of frames processed per second.

**Baselines.** We compare with representative methods at each supervision level: weakly-supervised, one-class, unsupervised, and zero-shot. For Holmes-VAD (Zhang et al., 2024), we quote the numbers reported in VERA (Ye et al., 2024) because the annotations for instruction tuning are not public.

Table 2: **Weakly-hard real-time assessment.** We evaluate the real-time capability of the proposed model by comparing accuracies on UCF-Crime (Sultani et al., 2018) and XD-Violence (Wu et al., 2020), together with performance metrics on 1280×720@30fps video. We provide the results of LAVAD[1] as a reference. All results are measured on an RTX 2000 Ada. The **best** values are highlighted.

| Method | UCF-Crime AUC (%) | XD-Violence AP (%) | XD-Violence AUC (%) | $p_{50}$ (sec, ↓) | $p_{95}$ (sec, ↓) | $p_{99}$ (sec, ↓) | $x^*/\xi$ (↓) | RTF (↓) | Throughput (frames/sec) | Speed up (times) |
|---|---|---|---|---|---|---|---|---|---|---|
| LAVAD (Zanella et al., 2024) | 80.28 | 62.01 | 85.36 | 153.108 | 171.643 | 184.496 | 1000/1000 | 119.77 | 0.19 | 1.0× |
| Flashback$_B$ (Ours) | 82.05 | 69.40 | 88.13 | **0.040** | **0.051** | **0.068** | 0/1000 | **0.04** | 125.29 | 675.5× |
| Flashback$_X$ (Ours) | 85.58 | 72.61 | **91.01** | 0.209 | 0.228 | 0.265 | 0/1000 | 0.21 | 43.82 | 236.2× |
| Flashback$_L$ (Ours) | **87.74** | **75.00** | 90.51 | 0.723 | 0.731 | 0.786 | 0/1000 | 0.73 | 12.17 | 65.6× |

Table 3: **Evaluation of reliability of descriptions.** Performance of Flashback versus the baselines and the official CUVA (method) (Du et al., 2024) on CUVA (dataset) `Description` task. Higher is better for all three metrics: BLEU, ROUGE–L, BLEURT. The **best** scores are highlighted.

| Method | VideoLM | BLEU (%) | ROUGE–L (%) | BLEURT (%) |
|---|---|---|---|---|
| | mPLUG-owl (Ye et al., 2023) | 0.55 | 12.58 | 40.66 |
| | Video-LLaMA (Zhang et al., 2023) | 0.60 | 13.15 | 40.55 |
| Baseline (Du et al., 2024) | PandaGPT (Su et al., 2023) | 0.66 | 13.33 | 38.23 |
| | Otter (Li et al., 2023a) | **1.07** | **15.19** | 29.92 |
| | Video-ChatGPT (Maaz et al., 2024) | 0.30 | 9.75 | 46.83 |
| CUVA (Du et al., 2024) | Video-ChatGPT (Maaz et al., 2024) | 0.55 | 14.35 | **47.10** |
| **Flashback (Ours)** | - | 0.36 | 10.63 | 41.86 |

**Implementation.** The pseudo-scene memory is generated once with `gpt-4o-2024-08-06` (OpenAI, 2024), costing \$181.43 and 76 hours for one million normal-anomalous pairs. The frozen cross-modal encoder is PerceptionEncoder (Bolya et al., 2025). We use `PE-Core-B` for Flashback$_B$, `PE-Core-L` for Flashback$_L$. Unless noted, we use Flashback$_L$ by default. We denote the switching model as Flashback$_X$. All experiments run on a single RTX 2000 Ada. Parameters are in Section C.2.

## 4.2 COMPARISON WITH STATE-OF-THE-ART METHODS

Flashback achieves the best results in every zero-shot setting, as summarized in Table 1. Flashback raises the state of the art by a large gain of XD AP over zero-shot LAVAD (Zanella et al., 2024) (62.01 vs. 75.13). The proposed method (XD AUC 90.54) also surpasses strong unsupervised (vs. 68.33), one-class (vs. 61.56), and several weakly-supervised baselines. Particularly, it exceeds the explainable VERA (Ye et al., 2024) on XD AP (70.54 vs. 75.13). To our knowledge, Flashback is the first VAD system that is simultaneously zero-shot, real-time, and able to return human-readable explanations.

## 4.3 WEAKLY-HARD REAL-TIME ASSESSMENT

Weakly-hard real-time (WHRT) constraints (Bernat et al., 2001) allow occasional misses as long as deadlines are met consistently, offering a bridge between theory and deployment. As shown in Table 2, for 1s segments LAVAD takes 153 seconds, while our method needs only 0.723 seconds. Across any $\xi = 1000$ segments no deadlines are missed ($x^* = 0$), and with RTF < 1 the WHRT condition holds. In addition, certainty shows a strong negative correlation with absolute error (Spearman (Kendall & Gibbons, 1990) $\rho = -0.58$; higher certainty corresponds to smaller error), indicating that likelihoods serve as error detectors and thus validate runtime model selection. Flashback$_X$ achieves accuracy close to Flashback$_L$ (UCF AUC 85.58 vs. 87.74) but delivers 4× higher throughput, 236× LAVAD's.

## 4.4 RELIABILITY OF RETRIEVED CAPTIONS

We test how well retrieved captions describe true anomalies using the `Description` task of CUVA (Du et al., 2024), which presents anomalous videos and asks for a free-form explanation. Details of the protocol and metrics are described in Section D.1.

**Baselines.** Following CUVA, we prompt state-of-the-art video-language models (VideoLMs) (Ye et al., 2023; Zhang et al., 2023; Su et al., 2023; Li et al., 2023a; Maaz et al., 2024) to generate anomaly descriptions. CUVA's own refinement is denoted CUVA in Table 3. Unlike these online methods, Flashback simply retrieves a pre-generated pseudo-caption.

**Results.** Table 3 shows that Flashback is competitive with large online VideoLMs: it surpasses Video-ChatGPT (Maaz et al., 2024) on ROUGE–L (Lin, 2004) (10.63 vs. 9.75) and outperforms

---

[1] Since the cleaning operation of LAVAD requires captioning the entire video before processing, we assume it occurs immediately. Moreover, the LLM used for captioning, `LLaMA-2-13b-chat`, demands large GPU memory, so inference is run on two RTX 6000 Ada GPUs instead of a single RTX 2000 Ada. Therefore, the reported numbers represent an *optimistic* estimate relative to realistic deployment.

Table 4: **Ablation study.** We report AUC on UCF-Crime (Sultani et al., 2018) and AUC/AP on XD-Violence (Wu et al., 2020) (a) Four disjoint subsets of 100k pseudo-captions are sampled from the 1M memory; results are shown as mean$_{\pm \text{std.dev.}}$. (d,e) RTF is reported with accuracy on a single RTX 2000 Ada. (f) AUC is also reported on naive anomaly datasets ShanghaiTech (Liu et al., 2018) and StreetScene (Ramachandra & Jones, 2020), with random and LAVAD (Zanella et al., 2024) as references. The **best** result is in bold; the main system is indicated with a gray shade .

(a) Stability and reproducibility of memory.

| Seed | UCF-Crime AUC (%) | XD-Violence AP (%) | XD-Violence AUC (%) |
|---|---|---|---|
| A | 87.27 | 74.29 | 90.15 |
| B | 87.35 | 73.90 | 90.00 |
| C | 86.06 | 74.55 | 90.15 |
| D | 86.33 | 74.66 | 90.16 |
| Overall | 86.75$_{\pm 0.65}$ | 74.35$_{\pm 0.34}$ | 90.12$_{\pm 0.08}$ |

(b) Effectiveness of repulsive prompting.

| Strategy | UCF-Crime AUC (%) | XD-Violence AP (%) | XD-Violence AUC (%) |
|---|---|---|---|
| ✗ | 74.98 | 71.01 | 87.08 |
| Lin. alg. op. | 81.56 | 64.98 | 83.04 |
| RP (keyword-only) | 81.24 | 72.20 | 88.42 |
| RP (template-only) | 82.12 | 72.21 | 88.82 |
| RP | **87.74** | **75.00** | **90.51** |

(c) Impact of the number of retrieved captions $K$.

| $K$ | UCF-Crime AUC (%) | XD-Violence AP (%) | XD-Violence AUC (%) |
|---|---|---|---|
| 1 | 82.00 | 73.55 | 88.46 |
| 5 | 85.66 | 74.84 | 90.19 |
| 10 | **87.74** | **75.00** | **90.51** |
| 20 | 86.84 | 75.08 | 90.71 |
| 40 | 86.31 | 74.86 | 90.73 |

(d) Effectiveness and efficiency of the size of memory.

| # Caption pairs | UCF-Crime AUC (%) | XD-Violence AP (%) | XD-Violence AUC (%) | RTF ($\downarrow$) |
|---|---|---|---|---|
| 10,000 | 85.46 | 71.87 | 88.55 | **0.70** |
| 50,000 | 87.07 | 74.25 | 89.98 | 0.71 |
| 100,000 | 86.75 | 74.35 | 90.12 | 0.71 |
| 500,000 | 87.32 | 74.94 | 90.46 | 0.72 |
| 1,000,000 | **87.74** | **75.00** | **90.51** | 0.73 |

(e) Effectiveness and efficiency of video sampling parameters.

| $T_{\text{segment}}$ (sec) | $T_{\text{sample}}$ (frames) | UCF-Crime AUC (%) | XD-Violence AP (%) | XD-Violence AUC (%) | RTF ($\downarrow$) |
|---|---|---|---|---|---|
| 1.0 | 1 | **87.82** | 72.78 | 89.96 | **0.11** |
| 1.0 | 8 | 87.74 | 75.00 | 90.51 | 0.73 |
| 1.0 | 16 | 87.72 | **75.13** | **90.54** | 1.40 |
| 0.5 | 8 | 85.56 | 73.81 | 90.52 | 1.45 |

(f) Expandability.

| Context prompt | ShanghaiTech AUC (%) | StreetScene AUC (%) |
|---|---|---|
| *References* | | |
| Random | 50.00 | 50.00 |
| LAVAD | 44.34 | 58.52 |
| Default | 58.56 | 64.12 |
| Domain-specific | **62.49** | **68.57** |

others on BLEURT (Sellam et al., 2020) (41.86 vs. 40.66, 40.55, 38.23, 29.92). This indicates that i) the pseudo-scene memory spans diverse anomalies, and ii) retrieval reliably matches each event.

## 4.5 ABLATION STUDY

Table 4 disentangles the effect of each design choice on accuracy: AUC for UCF-Crime (Sultani et al., 2018) (UCF), AUC and AP for XD-Violence (Wu et al., 2020) (XD).

**(a) Stability and reproducibility.** We randomly draw four disjoint subsets of 100k captions from the 1M-entry memory and retrain nothing. The frame-level AUC varies only $86.75 \pm 0.65$ on UCF and $90.12 \pm 0.08$ on XD, showing that performance does not depend on a particular subset.

**(b) Repulsive prompting (RP).** Removing RP drops UCF AUC from 87.74 to 74.98 and XD AP from 75.00 to 71.01, as caption embeddings collapse without it as illustrated in Figure 3. We split RP into two variants: *keyword-only*, inserting Normal/Anomalous tokens, and *template-only*, adding the wrapper. Each recovers part of the accuracy, but only the full RP restores the gain.

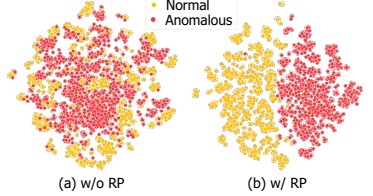

(a) w/o RP     (b) w/ RP

Figure 3: **T-SNE representation of text features.** We subsample 5,000 normal-anomalous caption pairs and visualize (a) before and (b) after applying repulsive prompting (RP). RP clearly separates the two groups.

We also test a geometric alternative that projects embeddings away from the anomaly axis (Section G.1). It gives a slight gain on UCF but reduces XD, while RP improves both, showing that input-level cues are more reliable than post-hoc vector shifts.

**(c) Top-$K$ captions.** We sweep $K \in \{1, 5, 10, 20, 40\}$. UCF AUC rises from 82.00 at $K = 1$ to 87.74 at $K = 10$ and then drops to 86.31 at $K = 40$; XD shows the same trend. With $K = 40$, many retrieved captions are loosely related to the segment. We therefore fix $K = 10$ for all main results.

**(d) Memory size and RTF.** Scaling the size of memory from 10k to 1M captions raises UCF AUC from 85.46 to 87.74 and XD AP from 71.87 to 75.00, while RTF changes only from 0.70 to 0.73. One can construct a larger memory for accuracy while keeping RTF virtually unchanged, as the dot product is much cheaper than model forward as shown in Table 6. It is worth noting that FP32 features for 1M captions take only 3.9 GiB of memory.

**(e) Segment length and sampling rate.** Sampling 8 frames in a 1.0s segment reaches the 2nd best score: 87.74 UCF AUC and 75.00 XD AP with fair RTF (0.73). Denser sampling ($T_{\text{sample}} = 16$) and shorter segments ($T_{\text{segment}} = 0.5$s) yield similar or lower XD AP (75.13, 73.81) yet do not speed up the pipeline, offering no benefit. Therefore, the gray row becomes our default.

**(f) Expandability.** ShanghaiTech (Liu et al., 2018) and StreetScene (Ramachandra & Jones, 2020) contain everyday anomalies such as "running in hallways," unlike crime-focused UCF and XD.

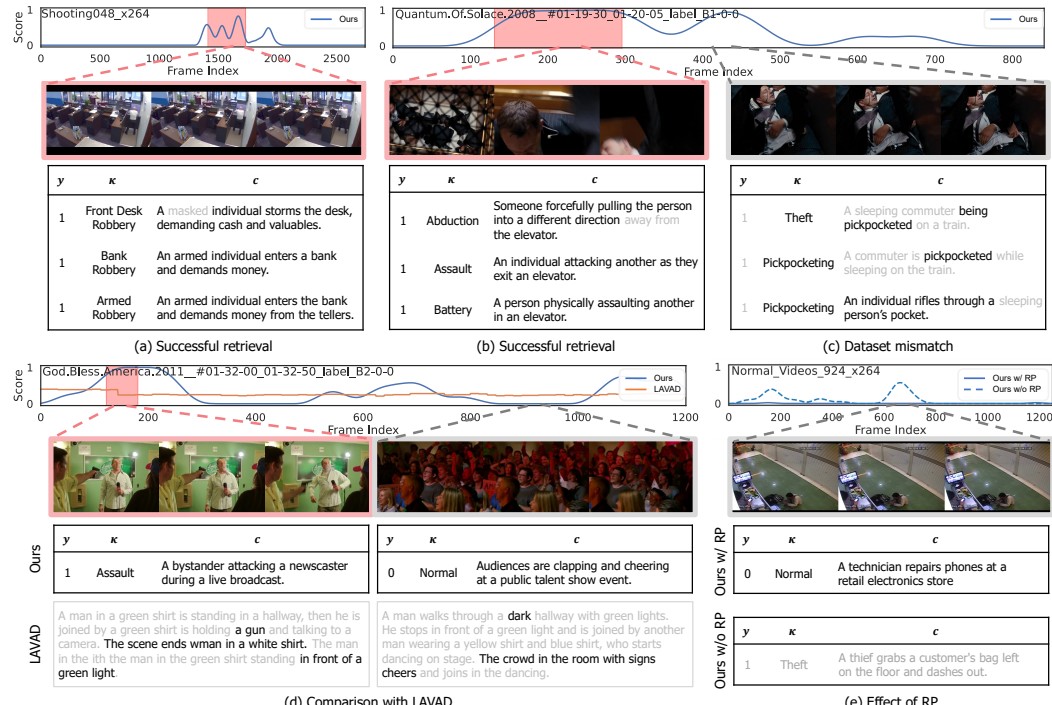

**Figure 4: Qualitative examples.** The plots show frame-wise anomaly curves, with red boxes marking ground-truth anomalous intervals on both the video strip and the plot. For selected segments, we list retrieved category-caption pairs $(\kappa, c)$ with anomaly flags $y$; black text denotes correct descriptions, gray incorrect ones. (a-b) Events are described precisely. (c) Label mismatch with XD-Violence. (d) Flashback vs. LAVAD curve contrast. (e) Repulsive prompting (RP) ablation yields false alarms.

Flashback adapts by adding domain-specific context (Section C) when building the pseudo-scene memory. LAVAD (Zanella et al., 2024) underperforms random on ShanghaiTech, highlighting its difficulty in handling naive anomalies, while Flashback reaches 62.49 AUC with Routine context.

**Effect of scaled anomaly penalization (SAP) and choice of $\alpha$.** To assess the impact of scaled anomaly penalization under realistic conditions, we merge UCF and XD into a single evaluation pool and sweep the scale factor $\alpha$ from 0.80 to 1.00 and show the result in Figure 6. Flashback achieves peak AUC at $\alpha \approx 0.95$, with stable performance across 0.90-1.0. We therefore fix $\alpha = 0.95$ in all experiments, reducing anomaly bias without per-dataset tuning.

### 4.6 Qualitative analysis

Figure 4 illustrates qualitative results of Flashback. Panels (a-b) illustrate how captions pre-generated without any video already exist in the memory, and retrieval selects them accurately at inference. Panel (c) shows a failure: the detector flags "Pickpocketing" as abnormal while XD treats it as normal. It is counted as a false positive though the explanation is plausible. Panel (d) compares LAVAD (Zanella et al., 2024), which yields flat curves, against Flashback, which shows clear slopes that simplify thresholding. Captions are concise and avoid malformed outputs. Finally, panel (e) shows that removing repulsive prompting (RP) increases false positives, especially on static normal shots.

## 5 Conclusions

The core contribution of Flashback is recasting video anomaly detection as retrieval over a pseudo-scene memory of text-only captions, eliminating reliance on video-conditioned models for caption generation. By building a pseudo-scene memory offline, it avoids online LLM calls; repulsive prompting and scaled anomaly penalization provide a clear margin between normal and anomalous captions; and runtime encoder selection balances speed and accuracy by allocating computation only where needed. Experiments show that Flashback outperforms prior zero-shot methods, rivals weakly-supervised methods, and consistently meets weakly-hard real-time constraints, while qualitative analyses confirm that retrieved captions align with human judgment and offer concise explanations.

## Ethics statement

We adhere to the ICLR Code of Ethics. No real anomalous videos were used for training; the pseudo-scene memory was generated entirely from language models, and videos were only used for evaluation on public benchmarks such as UCF-Crime and XD-Violence. We intend to release code and assets to support reproducibility, subject to approval from our organization, and in the meantime provide full experimental details to enable independent verification. While our system is designed for research and beneficial applications such as safety monitoring, it could be misused for intrusive surveillance or other unintended purposes. We emphasize that our contribution is methodological and encourage deployment strictly in transparent and ethically appropriate contexts.

## Reproducibility statement

We will release code, generated captions, and features once institutional approval is obtained. To ensure reproducibility even without our artifacts, we provide extensive details: all hyperparameters and models are listed in Section 4.1 and Section C.2, while Section 3, Section C, and Section D describe the reproduction procedure and mathematical derivations. All datasets used for evaluation (Sultani et al., 2018; Wu et al., 2020; Liu et al., 2018; Ramachandra & Jones, 2020; Du et al., 2024) are publicly available, and the metrics are defined in Section 4.1. The LLM (OpenAI, 2024) is accessible via API, and the video-text encoder, PerceptionEncoder (Bolya et al., 2025), is publicly released. As shown in Table 4 (a), performance remains stable across random pseudo-caption samples, suggesting that independent generation yields comparable results. Generating 1M captions required approximately $180 and 76 hours, which we consider feasible for academic researchers, and Table 4 (d) shows that even 100k captions provide reasonable performance. Flashback involves no stochastic operations after pseudo-caption generation; nevertheless, we fix all random seeds (e.g., NumPy, PyTorch) to 42 for completeness.

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

## A  NOTATION

| Symbol | Meaning |
|---|---|
| $V = \{I_t\}_{t \leq 1}$ | Input video as a sequence of frames |
| $r$ | The sampling rate in frames per second |
| $V_s$ | The $s$-th video segment |
| $T$ | The number of frames of a video |
| $T_{\text{sample}}$ | The number of frames sampled per segment |
| $T_{\text{segment}}$ | Segment length in seconds |
| $\tau_s$ | The wall-clock time to process the $s$-th segment |
| $\xi$ | The number of consecutive segments for WHRT window |
| $f_{\text{text}}$ | Text encoder (frozen) |
| $f_{\text{video}}$ | Video encoder (frozen) |
| $D$ | The dimensionality of the joint video-text embedding space |
| $\mathcal{C}_{\text{N}}, \mathcal{C}_{\text{A}}$ | Sequences of pseudo-captions (normal, anomalous) |
| $\mathcal{K}_{\text{N}}, \mathcal{K}_{\text{A}}$ | Sequences of action categories for pseudo-captions |
| $\mathcal{Y}$ | Binary anomaly labels for all pseudo-captions |
| $\mathcal{Z}_{\text{N}}, \mathcal{Z}_{\text{A}} \subseteq \mathbb{R}^D$ | Text embeddings of normal and anomalous captions |
| $\alpha > 0$ | Scale factor for anomalous captions in SAP |
| $\mathbf{t}_j \in \mathbb{R}^D$ | Embedding of the $j$-th pseudo-caption |
| $\tilde{\mathbf{t}}_j \in \mathbb{R}^D$ | Scaled anomalous embedding after SAP |
| $\mathbf{v}_s \in \mathbb{R}^D$ | Video embedding of segment $V_s$ |
| $\sigma_{s,j} \in \mathbb{R}$ | Dot product similarity between $\mathbf{v}_s$ and $\mathbf{t}_j$ |
| $K$ | Number of captions to retrieve for each segment |
| $\mathbf{k}_s^* \in \mathbb{N}^K$ | Indices of top-$K$ retrieved captions for segment $s$ |
| $\mathbf{w}_s^* \in (0,1)^K$ | Softmax-normalized retrieval weights |
| $\mathbf{c}_s^* \in \mathcal{C}^K$ | Captions retrieved for the $s$-th segment |
| $\mathbf{y}_s^* \in \{0,1\}^K$ | Anomaly flags retrieved for the $s$-th segment |
| $A_s \in [0,1]$ | Segment-level anomaly score |
| $p_t, \tilde{p}_t \in [0,1]$ | Frame-level anomaly score at frame $t$ before/after smoothing |
| $\mathbf{F} \in \mathbb{R}^{2 \times 2}$ | Transition matrix of Kalman filter |
| $\mathbf{H} \in \mathbb{R}^{1 \times 2}$ | Observation matrix of Kalman filter |
| $\mathbf{Q} \in \mathbb{R}^{2 \times 2}, \succ \mathbf{0}$ | Process noise matrix of Kalman filter |
| $R > 0$ | Observation noise of Kalman filter |
| $\tilde{a}_s \in \mathbb{R}$ | Innovation of Kalman filter for segment $s$ |
| $S_s > 0$ | Innovation variance of Kalman filter for segment $s$ |
| $H$ | Number of latest certainties for deciding adaptive thresholds |
| $\lambda_s \leq 0$ | Certainty for segment $s$ (Higher the more certain) |

Table 5: Notation used throughout the paper.

## B  LLM USAGE

We used large language models in two ways. ChatGPT was used for language polishing; all technical content, experiments, and results were designed and verified by the authors, with code assistance from AI coding tools (VSCode Copilot and Cursor). Second, in the method itself, we employed an LLM to generate pseudo-scene captions offline for constructing the memory used in anomaly detection. The authors take full responsibility for all content.

## C  FURTHER DETAILS OF FLASHBACK

This section provides additional details that clarify how Flashback operates in practice and complements the high-level description in the main paper. We document the exact procedure for constructing the pseudo-scene memory with an LLM, including prompt design, batching strategy, and the observed variability across random seeds. We also present the domain-specific context prompts used for extendability experiments and list all hyper-parameters required for full reproducibility, including

```
class Completions(SyncAPIResource):
    @overload
    def create(
        self,
        *,
        ...
        seed: Optional[int] | NotGiven = NOT_GIVEN,
        ...
    ) -> ChatCompletion:
        """
        ...
        Args:
            ...
            seed: This feature is in Beta. If specified, our system will make a best effort to
                sample deterministically, such that repeated requests with the same `seed` and
                parameters should return the same result. Determinism is not guaranteed, and you
                should refer to the `system_fingerprint` response parameter to monitor changes
                in the backend.
        ...
        """
        ...
```

Figure 5: **Docstring for `'seed'` argument of OpenAI Python API `1.75.0`.**

those for the runtime model selection and the Kalman filter. Together, these details offer a transparent view of the components that support our zero-shot, real-time anomaly detection system.

## C.1 IMPLEMENTATION DETAILS OF THE LLM CALL

We instruct GPT-4o (OpenAI, 2024) to answer in valid JSON. Because every brace and quote must close, a truncated response is unusable and must be regenerated. To keep outputs short enough, the user message simply asks: "Generate 100 example scene descriptions." The two system messages, the context prompt $P_C$ and the format prompt $P_F$, remain fixed, while the 100-caption request is repeated 10,000 times with new seeds, producing one million pseudo-captions.

The OpenAI Python API explicitly warns that full determinism is not guaranteed, even with a fixed seed as illustrated in Figure 5. Nonetheless, Table 4 (a) shows that different caption batches yield nearly identical accuracy, confirming that Flashback is robust to these small variations.

**Domain-specific context prompt $P'_C$.** To demonstrate expandability in Table 4 (f), we generate pseudo-captions by prompting the LLM with additional context so that even everyday minor rule violations are flagged as anomalies. The exact domain-specific context prompts for ShanghaiTech (Liu et al., 2018) and StreetScene (Ramachandra & Jones, 2020) are shown below. They specify the setting of a university campus or a traffic monitoring and provide examples of anomalies such as someone running in a hallway, or provide a role of a vigilant police officer. We use the same format prompt $P_F$.

> for ShanghaiTech: `You are solving the video anomaly detection (VAD) problem in a fancy way. As you know, anomalous events are rare but their categories are diverse. You have to generate example scene descriptions both for the anomalous events and normal events. We will use these descriptions to decide if given video clips contain anomalous events by choosing one of the descriptions having the top similarity measured by a multi-modal retrieval model. The descriptions should be short and concise. The anomalous events should focus on routine irregularities that might occur in a university campus (e.g., someone running in a hallway, climbing over a fence, leaving belongings unattended), not severe incidents like crimes. The entire response should be in the provided json format.`
>
> for StreetScene: `You are a vigilant police officer monitoring CCTV footage. Your job is to spot and catch citizens or vehicles that violate traffic rules. You are solving the video anomaly detection (VAD) problem in a fancy way: anomalous events are rare but their categories are diverse. You must generate example scene descriptions for both anomalous traffic violations and normal routine behaviors. We will use these descriptions to decide if given`

```
              video clips contain anomalous events by choosing one of
              the descriptions having the top similarity measured by a
              multi-modal retrieval model. The descriptions should be
              short and concise. The entire response should be in the
              provided json format.
```

## C.2 HYPER-PARAMETERS

Unless noted, we use Flashback$_\text{L}$ by default and use $T_\text{segment} = 1$s, $T_\text{sample} = 8$ frames, input resolution $336\times336$, Gaussian kernel with 100-frame width and $\sigma = 0.5$. Categories for normal captions $\mathcal{K}_\text{N}$ are empty strings. We set the number of captions for retrieval $K$ as 10. For adaptive retriever switching, we set $\mathbf{Q} = \text{diag}(1, 10^{-3})$, $R = 1.5$, $\tau = 5$, and initialize with the state vector $\mathbf{x}_0 = \mathbf{0}$, and the predicted estimate covariance $\mathbf{P}_0 = \text{diag}(0.1, 0.1)$.

# D KALMAN-BASED LIKELIHOOD FOR SEGMENT SCORE CERTAINTY

**Model.** Let the latent state be $\mathbf{x}_s \in \mathbb{R}^2$ and the observed segment score be $a_s \triangleq A_s \in \mathbb{R}$. Use the linear-Gaussian state space model

$$\mathbf{x}_s = \mathbf{F}\,\mathbf{x}_{s-1} + \mathbf{w}_s, \quad \mathbf{w}_s \sim \mathcal{N}(\mathbf{0}, \mathbf{Q}), \qquad a_s = \mathbf{H}\,\mathbf{x}_s + v_s, \quad v_s \sim \mathcal{N}(0, R),$$

with

$$\mathbf{F} = \begin{bmatrix} 1 & \Delta t \\ 0 & \phi \end{bmatrix}, \quad \mathbf{H} = \begin{bmatrix} 1 & 0 \end{bmatrix}, \quad \phi = e^{-\Delta t/\tau},$$

and hyperparameters $\mathbf{Q} \succeq 0$, $R > 0$, $\Delta t > 0$, $\tau > 0$.

**Kalman recursions.** Given the filtered mean and covariance $(\hat{\mathbf{x}}_{s-1}, \mathbf{P}_{s-1})$, compute the prediction:

$$\hat{\mathbf{x}}_{s|s-1} = \mathbf{F}\,\hat{\mathbf{x}}_{s-1}, \qquad \mathbf{P}_{s|s-1} = \mathbf{F}\,\mathbf{P}_{s-1}\,\mathbf{F}^\top + \mathbf{Q}.$$

Form the innovation (a.k.a. residual) and its variance:

$$\tilde{a}_s \triangleq a_s - \mathbf{H}\hat{\mathbf{x}}_{s|s-1}, \qquad S_s \triangleq \mathbf{H}\mathbf{P}_{s|s-1}\mathbf{H}^\top + R.$$

Update the filtered state:

$$K_s = \mathbf{P}_{s|s-1}\mathbf{H}^\top S_s^{-1}, \qquad \hat{\mathbf{x}}_s = \hat{\mathbf{x}}_{s|s-1} + K_s\tilde{a}_s, \qquad \mathbf{P}_s = (\mathbf{I} - K_s\mathbf{H})\,\mathbf{P}_{s|s-1}.$$

**Innovation log-likelihood (certainty).** Under the linear-Gaussian model, the conditional density of $a_s$ given past observations equals

$$p(a_s \mid a_{1:s-1}) = \mathcal{N}\big(\mathbf{H}\hat{\mathbf{x}}_{s|s-1}, S_s\big).$$

The log-likelihood contribution for step $s$ follows:

$$\lambda_s \triangleq \log p(a_s \mid a_{1:s-1}) = -\frac{1}{2}\left(\frac{\tilde{a}_s^2}{S_s} + \log S_s + \log 2\pi\right), \quad \lambda_s \in (-\infty, 0].$$

Because $a_s \in \mathbb{R}$, $S_s$ is scalar, so evaluation costs $O(1)$ per segment.

**Hyperparameters and interpretation.** $\tau$ controls temporal persistence via $\phi = e^{-\Delta t/\tau}$. $\mathbf{Q}$ controls state drift; $R$ controls observed score noise. Smaller $R$ or larger $\mathbf{Q}$ increases sensitivity to short-term deviations, which raises $|\lambda_s|$ on surprising segments.

**Decision rule for runtime model selection.** Maintain a rolling buffer of recent $\{\lambda_{s'}\}$. Compute empirical quantiles $q_\text{bottom}$ and $q_\text{top}$. If $\lambda_s < q_\text{bottom}$, switch to the larger encoder for refinement. If $\lambda_s > q_\text{top}$, return to the smaller encoder.

## D.1 METRICS AND PROTOCOL FOR EVALUATING RELIABILITY OF CAPTIONS

**Metrics.** BLEU (Papineni et al., 2002), ROUGE–L (Lin, 2004), and BLEURT (Sellam et al., 2020) evaluate different facets of caption quality. **BLEU** (Bilingual Evaluation Understudy) is an $n$-gram *precision* metric: it counts 1-4-gram overlaps between the candidate and the reference and applies a brevity penalty, so high BLEU means the generated caption repeats the reference's key content

words in roughly the same order. **ROUGE–L** (Recall-Oriented Understudy for Gisting Evaluation, longest-common-subsequence variant) measures *recall*; it finds the longest sequence of words that appear in the same order (not necessarily contiguously) in both sentences. Because it tolerates gaps, ROUGE–L rewards paraphrases that keep the gist even when the wording differs. **BLEURT** (BERT-based Learned Evaluation with Representations from Transformers) embeds both sentences with a fine-tuned BERT model and predicts a scalar similarity trained to match human judgments, allowing it to credit semantic equivalence even with little lexical overlap (e.g., "wielding a gun" vs. "pointing a firearm").

Thus, BLEU highlights exact keyword preservation, ROUGE–L captures sequence-level coverage of reference content, and BLEURT provides a semantics-aware view; consistent gains across all three indicate captions that match both the surface terms and the intended meaning of the ground-truth descriptions.

**From segment to video description.** CUVA expects one caption per video, whereas Flashback works at the segment level. We therefore (i) embed all segments and captions as explained in Section 3, (ii) pick the segment whose $K$ caption-video dot-products sum to the largest value, and (iii) join its $K$ retrieved captions with '; ' to form a single video-level description.

Table 6: **Latency distribution**. Latency (ms) and relative portion for processing a 1-second segment. Forward pass dominates the cost, while retrieval is negligible.

| Phase | Latency (ms) | Portion (%) |
|---|---|---|
| decode | 9.7 | 1.3 |
| transform | 15.5 | 2.1 |
| fetch | 1.8 | 0.3 |
| forward | 678.8 | 93.5 |
| retrieve | 19.5 | 2.7 |
| scoring | 0.5 | 0.1 |

## E    LIMITATIONS

The pseudo-scene memory assigns fixed anomaly labels to captions, even though some behaviors depend on context, which matters in certain real-world deployments. However, this affects UCF-Crime (Sultani et al., 2018) and XD-Violence (Wu et al., 2020) only mildly because their anomaly definitions are unambiguous. The system explains predictions only through retrieved captions, which restricts the depth of reasoning but still produces clean and relevant rationales at real-time speed. Flashback also inherits biases present in LLM-generated captions, which can be significant in rare or culturally sensitive scenes. However, the current benchmarks contain only rigid anomalies. Thus, we keep this risk limited. Finally, the method analyzes short video segments without modeling long-range temporal structure or audio cues, yet it still outperforms methods that use such signals, which suggests that this limitation is not critical for the benchmark datasets.

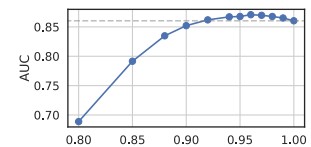

Figure 6: **AUC vs. scale factor $\alpha$.** A mild reduction ($\alpha \approx 0.95$) yields favorable AUC, confirming that scaled anomaly penalization is effective without fine-tuning.

## F    FUTURE WORK

Future work will extend Flashback along three main directions. First, our current system reasons over short segments with a fixed temporal horizon. Extending it with long-range temporal reasoning, while still meeting strict real-time latency constraints, would allow Flashback to capture slowly evolving or staged events, and to integrate auxiliary modalities such as audio and spatial cues into the pseudo-scene memory for more complex, multi-actor scenarios. Second, on the representation side, we plan to study lightweight adaptation and automated debiasing of caption embeddings. For example, learning context-dependent scaling between normal and anomalous captions or refining context-aware labeling. This will make margin between normal and anomalous text representations better reflects environment-dependent notions of abnormality. Third, we are interested in compact on-device aggregation and reasoning modules that can operate on top of retrieved captions, supporting intent-oriented queries and online refinement of the pseudo-scene memory without sacrificing strict real-time operation. Taken together, these directions aim to turn Flashback into a more robust, temporally aware, and interactive zero-shot anomaly detection framework.

## G    COMPARED METHOD DETAILS

This section summarizes the compared methods used in our ablations, including a geometric post-hoc baseline introduced only as a rough point of comparison. This baseline tests whether a simple vector shift could substitute for repulsive prompting, and its inconsistent behavior confirms that it cannot. We outline how the anomaly axis and deflation step are constructed for this evaluation.

### G.1 GEOMETRIC BASELINE FOR THE REPULSIVE-PROMPTING ABLATION

Row 2 of Table 4 (b) tests whether a purely geometric, post-hoc correction can substitute for repulsive prompting (RP). We first form the *anomaly axis*

$$\Delta \;=\; \frac{\boldsymbol{\mu}_{\mathrm{A}} - \boldsymbol{\mu}_{\mathrm{N}}}{\|\boldsymbol{\mu}_{\mathrm{A}} - \boldsymbol{\mu}_{\mathrm{N}}\|}, \qquad \boldsymbol{\mu}_v \;=\; \mathbb{E}_s[\mathbf{v}_s], \tag{1}$$

where $\boldsymbol{\mu}_{\mathrm{N}}$ and $\boldsymbol{\mu}_{\mathrm{A}}$ are the mean caption embeddings for normal and anomalous pseudo-captions, and $\boldsymbol{\mu}_v$ is the video-level mean of segment embeddings. Each segment feature is then "deflated" along that axis:

$$\mathbf{v}'_s \;=\; \mathbf{v}_s - \left(\Delta^\top \boldsymbol{\mu}_v\right) \Delta. \tag{2}$$

The intuition is that $\boldsymbol{\mu}_v$ should be anomaly-neutral; subtracting its projection onto $\Delta$ should damp bias toward anomalous directions.

The tweak helps on UCF-Crime, raising frame-AUC from 74.98 to 81.56, but hurts XD-Violence, dropping AP from 71.01 to 64.98. These mixed results confirm that carefully crafted input tokens (RP) yield a more stable separation than a global vector shift applied after the fact, and they do so without any extra training or parameters.

## H FURTHER RESULTS

This section presents additional qualitative results that illustrate how Flashback behaves on diverse videos beyond those shown in the main paper. The examples highlight the captions retrieved from the pseudo-scene memory and the resulting anomaly score patterns, offering a clearer view of how the system explains and localizes events in practice.

### H.1 QUALITATIVE EXAMPLES

To illustrate how Flashback behaves in practice, we show qualitative results in Figure 7 and Figure 8. We sample four videos each from UCF-Crime Sultani et al. (2018) and XD-Violence Wu et al. (2020). The pseudo-scene memory produces captions for a diverse set of situations, e.g., `Robbery`, `Explosion`, `Riot`, even though it was generated without target-domain videos, which qualitatively supports the zero-shot claim. Retrieving captions with anomaly flags from this memory yields anomaly scores that correlate well with the ground-truth events. Redundant phrasing is common among normal captions, whereas anomalous captions around a score peak often list several distinct but related event categories (`Hijacking`, `Mugging`, `Robbery`). Although a single caption may miss minor details, the top-$K$ captions together provide a comprehensive textual explanation of the ongoing scene.

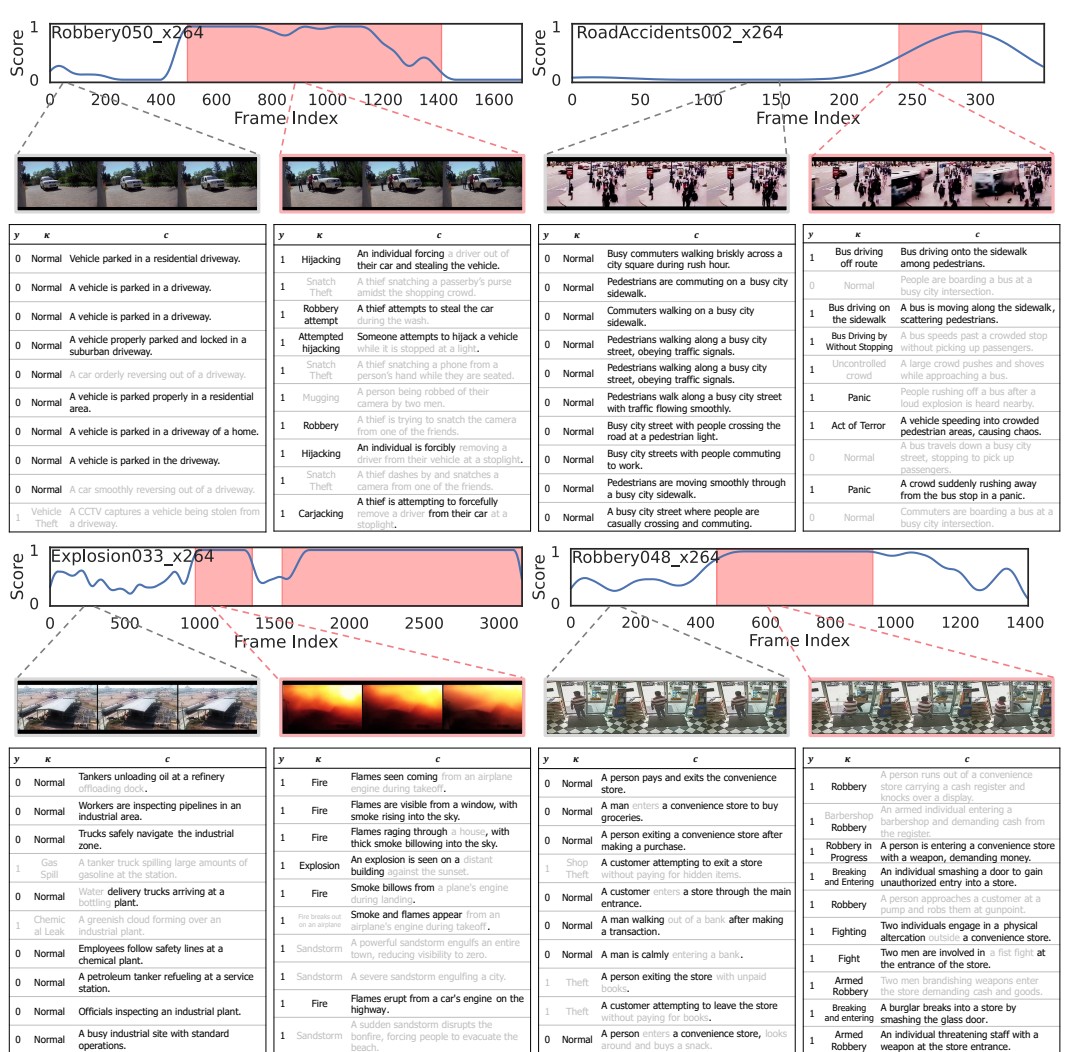

Figure 7: **Qualitative examples on UCF-Crime.** Red boxes on both the video strip and the plot mark ground-truth anomalous intervals . For selected segments, we list the retrieved category-caption pairs $(\kappa, c)$ and their anomaly flags $y$ in the descending order of similarities. Black text denotes a correct description, gray text an incorrect one.

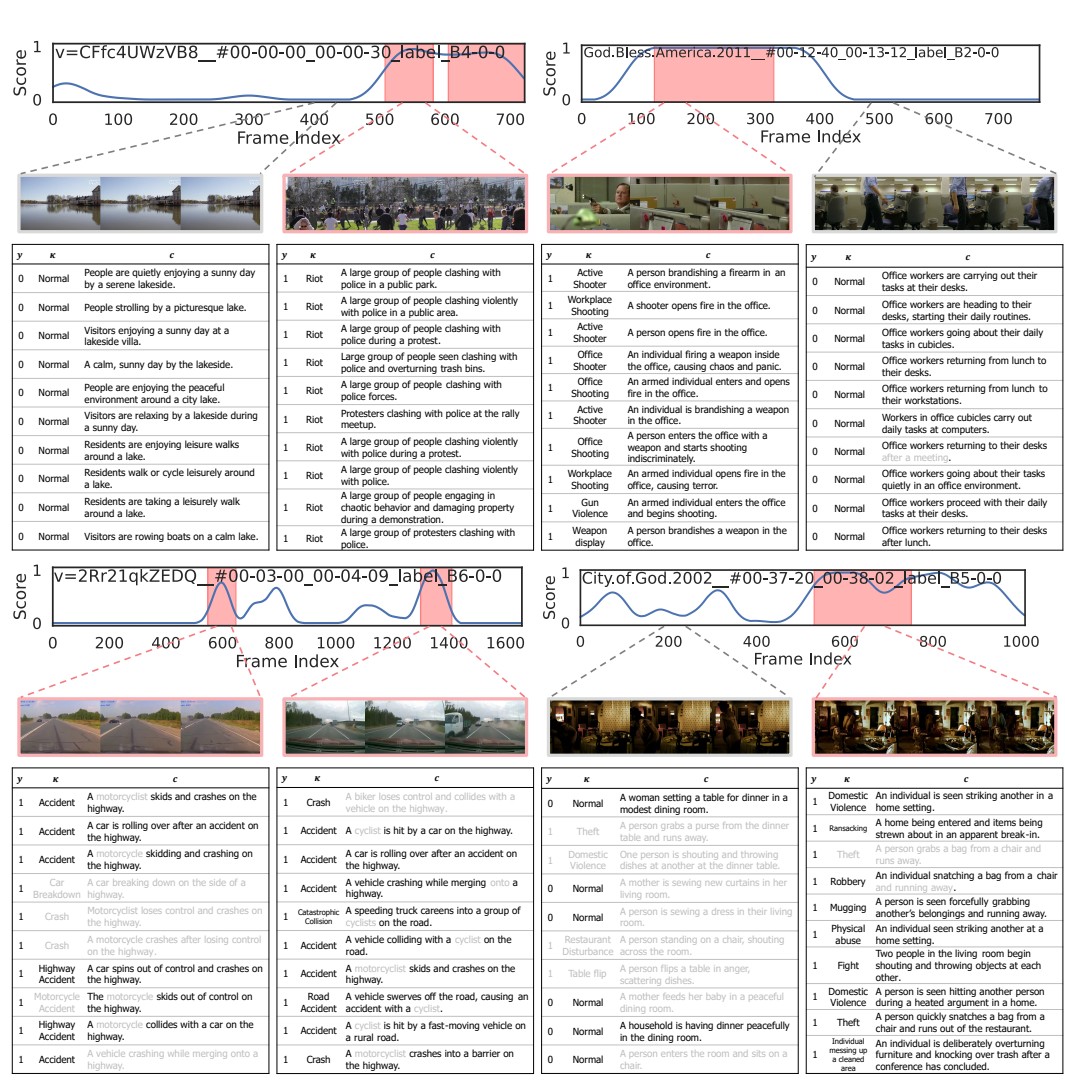

Figure 8: **Qualitative examples on XD-Violence.** The layout and notation match Figure 7.

