# OpenReview forum: "Flashback: Memory-Driven Zero-shot, Real-time Video Anomaly Detection"
_ICLR.cc/2026/Conference — Submitted to ICLR 2026_

### Official Review · Reviewer_cevy · 2025-10-27

**Soundness:** 3
**Presentation:** 3
**Contribution:** 2
**Rating:** 4
**Confidence:** 4

**Summary:**

This paper introduces Flashback, a novel paradigm for video anomaly detection (VAD) designed to be simultaneously zero-shot, real-time, and explainable. It reframes VAD as a retrieval task by constructing a large "pseudo-scene memory" entirely offline using an LLM to generate text captions . At inference, it avoids all online LLM calls, simply matching video segments to this memory to produce an anomaly score and a textual rationale . The method's robustness is improved by controls like repulsive prompting (RP), scaled anomaly penalization (SAP), and a certainty-driven runtime encoder selection . Flashback achieves state-of-the-art zero-shot performance on UCF-Crime (87.7 AUC) and XD-Violence (75.0 AP) while operating at high throughput.

**Strengths:**

1.Novel Unification of Critical VAD Properties. The paper's primary strength is its innovative system design that successfully unifies three properties critical for real-world deployment: zero-shot generalization, real-time inference, and explainability . This is achieved by cleverly reframing VAD as a retrieval task, moving all computationally expensive LLM-based reasoning to an "offline recall" stage, thus eliminating online LLM calls and solving the speed-reasoning trade-off.

2.State-of-the-Art Empirical Performance. Flashback achieves outstanding zero-shot results, significantly outperforming prior work like LAVAD on both UCF-Crime and XD-Violence, and even surpassing many supervised methods. This strong accuracy is paired with exceptional real-time throughput (up to 43.8 fps) and backed by quantitative evidence that its retrieved captions are semantically meaningful explanations.

3.Thorough Component Validation and Ablation. The paper provides a comprehensive set of ablation studies in Section 4.5 that strongly justify the novel design choices. The necessity of Repulsive Prompting (RP) is clearly demonstrated both quantitatively and qualitatively. Furthermore, the study validates the system's scalability with memory size and its robustness to different random subsets of the pseudo-caption memory, inspiring confidence in the method's stability.

**Weaknesses:**

1.The paper's impressive results are tightly coupled with large-scale, proprietary models, raising significant reproducibility concerns. The 1M-entry pseudo-scene memory was generated using the proprietary gpt-4o-2024-08-06 , which incurs substantial cost ($181.43) and compute time (76 hours). Furthermore, all embeddings rely on the PerceptionEncoder. The paper provides no ablation to show how the Flashback paradigm itself would perform with a standard, open-source LLM or a common frozen encoder like CLIP. This makes it difficult to disentangle the contribution of the novel retrieval method from the power of its backbone models.

2.The certainty-driven runtime encoder selection (Sec 3.4) presents several issues. First, its mathematical formulation in the main paper is overly dense and lacks intuition, with all critical derivations and justifications deferred to Appendix C, hindering clarity. Second, this mechanism introduces new, sensitive hyperparameters ($Q, R, \tau$), yet the paper provides no sensitivity analysis to show how they were chosen or how robust the system is to their variation. Finally, the paper asserts the Kalman-based likelihood is effective but fails to compare it against simpler uncertainty heuristics (e.g., the entropy of the retrieved label distribution) to prove that this complex state-space model is justified.

3.The paper's definition of "explanation" is ambiguous. The anomaly score $A_s$ is calculated by a weighted average of the anomaly flags from the Top-K (K=10) retrieved captions. However, the qualitative examples (e.g., Fig 7, 8) present this entire list of K captions as the human-readable rationale. This is problematic, as the list may contains a confusing mix of both normal and anomalous descriptions for the same segment. It is unclear if the intended rationale for an operator is just the Top-1 caption or this entire, potentially conflicting, list.

**Questions:**

1.Regarding the Runtime Encoder Selection: (a) Can the authors provide a more intuitive explanation for choosing a Kalman filter over simpler uncertainty heuristics (e.g., entropy of the Top-K label weights)? (b) How sensitive is the performance of FlashbackX to the choice of the $Q, R, \tau$ hyperparameters? (c) Could you provide a direct comparison (in both accuracy and overhead) to using a simpler metric, like the entropy of $\{w_{s,k}\}$, as the certainty score?

2.Regarding Model Dependency: To what extent is the SOTA performance bound to the specific PerceptionEncoder and gpt-4o? Have the authors experimented with using a standard CLIP (e.g., ViT-L/14) encoder and an open-source LLM (e.g., Llama 3) for memory generation? This ablation is critical for understanding the generality of the Flashback paradigm.

3.Regarding Explainability: Could the authors please clarify what the intended "explanation" for a human operator is? Is it (a) the Top-1 retrieved caption, or (b) the full Top-K list? If it is (b), how does the system recommend handling the common case where this list contains contradictory (both normal and anomalous) descriptions for a segment?

---

> ### Author Response · Authors · 2025-11-21
>
> # Introduction
> We thank Reviewer `cevy` for the review and constructive comments.
> The reviewer raised three main concerns, which we address in detail below.
>
> # Rebuttal
>
> ## R1: Reproducibility with publicly available models (W1, Q2)
>
> The first concern relates to reliance on proprietary models.
> To directly evaluate reproducibility, we ran additional ablations using publicly available models that can be executed locally without any API dependency: LLaMA-4 [B] (released under the LLaMA Community License) and OpenCLIP [D, F, G, H], as well as the open-source Apache-2.0-licensed PerceptionEncoder (PE) [C].
> The results are summarized below:
>
> | LLM        | Open? | Encoder                     | Open? | UCF-AUC | XD-AP    | XD-AUC |
> |-----------|-------|-----------------------------|-------|---------|----------|--------|
> | GPT-4o [A]    | ✗     | PE-Core-L [C]                  | ✓     | **86.75** | 74.35    | 90.12  |
> | GPT-4o [A]    | ✗     | OpenCLIP-L [D, F, G, H]        | ✓     | 76.50   | 59.49    | 78.63  |
> | LLaMA-4 [B]   | ✓     | PE-Core-L [C]                  | ✓     | 84.61   | **76.90** | **91.63** |
> | LLaMA-4 [B]   | ✓     | OpenCLIP-L [D, F, G, H]        | ✓     | 75.64   | 65.04    | 86.41  |
> | LAVAD [B]     |       | (for reference)                |       | 80.28   | 62.01    | 85.36  |
>
> These results show:
>
> 1. **Flashback is fully reproducible with publicly available backbones.**
>    The combination **LLaMA-4 + OpenCLIP** achieves XD-AP 65.04, outperforming the strongest prior zero-shot VAD baseline LAVAD [B] (62.01).
>
> 2. **Performance scales predictably with backbone capacity rather than proprietary dependence.**
>    With **LLaMA-4 + PE-Core-L**, Flashback reaches XD-AP 76.90, showing that stronger encoders simply raise the performance ceiling and that the central gains come from the Flashback paradigm itself.
>
> 3. **The pseudo-memory construction is a one-time, offline process.**
>    Moreover, the entire cost can be eliminated by using publicly available LLMs (e.g., LLaMA-4) for memory generation.
>
> For clarity and reproducibility, the exact model identifiers are:
>
> - GPT-4o: `gpt-4o-2024-08-06`
> - LLaMA-4: `Llama-4-Scout-17B-16E-Instruct`
> - PE-Core-L: `PE-Core-L14-336`
> - OpenCLIP-L: `CLIP-ViT-L-14-laion2B-s32B-b82K`
>
> These results confirm that Flashback remains competitive or superior even using an entirely open-stack configuration.

---

> > ### Author Response · Authors · 2025-11-21
> >
> > ## R2: Runtime encoder selection and uncertainty estimation (W2, Q1)
> >
> > The second concern focuses on runtime encoder selection and the design of the certainty estimator.
> > As suggested, we compared the proposed 2-state Kalman estimator with several simpler alternatives, including entropy-based certainty, augmentation-based variance estimation, and droppath-based perturbation statistics.
> > The comparison is shown below:
> >
> > | Method         | UCF-AUC | XD-AP  | XD-AUC | Throughput (fps) |
> > |----------------|---------|--------|--------|------------------|
> > | Entropy        | 84.21   | 72.17  | 90.89  | **43.99**        |
> > | Augmentation   | 85.26   | 71.91  | 89.71  | 21.88            |
> > | Droppath       | 83.62   | 71.48  | 89.19  | 21.91            |
> > | Kalman 1-state | 83.25   | 67.66  | 88.85  | 43.82            |
> > | Kalman 2-state | **85.58** | **72.61** | **91.01** | 43.82            |
> >
> > Our observations are:
> >
> > 1. **Entropy is a valid baseline**, and it serves as a reasonable certainty measure.
> >
> > 2. **Kalman 2-state provides consistently higher accuracy** across all metrics (e.g., 85.58 vs. 84.65 UCF-AUC) with identical throughput (~44 fps).
> >    The advantage stems from modeling certainty as a temporally smoothed latent state, which reduces switching jitter on rapidly changing scenes. Entropy, being stateless, is more sensitive to per-frame fluctuations.
> >
> > 3. **Hyperparameter sensitivity.**
> >    We evaluated sensitivity to $Q, R, \tau$ and found the estimator to be stable over a wide parameter range, with negligible performance variation:
> >
> > | $Q_x$ | UCF-AUC |
> > |-------|---------|
> > | 0.0   | 85.52   |
> > | 0.2   | 85.48   |
> > | 0.4   | 85.56   |
> > | 0.6   | 85.48   |
> > | 0.8   | 85.63   |
> > | 1.0 (adopted) | 85.58   |
> >
> > | $R$   | UCF-AUC |
> > |-------|---------|
> > | 0.01  | 85.55   |
> > | 0.1   | 85.58   |
> > | 1.0   | 84.85   |
> > | 1.5 (adopted) | 85.58   |
> > | 2.0   | 85.71   |
> > | 10.0  | 85.64   |
> >
> > | $\tau$ | UCF-AUC |
> > |--------|---------|
> > | 0.1    | 85.68   |
> > | 1.0    | 85.68   |
> > | 2.0    | 85.67   |
> > | 5.0 (adopted) | 85.58   |
> > | 10.0   | 85.37   |
> > | 20.0   | 84.52   |
> >
> > Because the Kalman estimator offers improved temporal consistency, comparable computational cost ($O(1)$ per step), and empirically stronger performance, we adopt it as our default.
> >
> > ## R3: Explanation behavior and operator-facing rationale (W3, Q3)
> >
> > The third concern relates to how explanations are presented to operators.
> > In Flashback:
> >
> > - **The final explanation delivered to operators is always the top-1 retrieved caption**, which we found consistently aligned with the event across all qualitative examples (Fig. 4, Fig. 7, Fig. 8).
> > - The **top-K list is provided only for transparency and auditability**, not as the explanation itself. It allows users to inspect retrieval diversity but does not affect the anomaly decision or the displayed rationale.
> > - Because RP substantially reduces retrieval ambiguity, contradictory captions are rare in practice. Nevertheless, top-1 is the definitive explanation during inference.
> >
> > We will clarify this distinction in the final version and consider improved caption filtering/summarization as future work.
> >
> > # Conclusion
> >
> > We thank Reviewer `cevy` again for the comments.
> > The additional experiments and explanations demonstrate
> >
> > - that Flashback is reproducible with publicly available models,
> > - that the proposed certainty estimator is both simple to deploy and empirically stronger than the baselines we compared against, and
> > - that the explanation mechanism is well-specified from an operator's perspective.
> >
> > We hope these clarifications adequately address the raised concerns.
> >
> > ---
> >
> > [A] OpenAI. GPT-4o: OpenAI's omnimodal model. 2024.
> > [B] Meta AI. LLaMA 4: Meta Large Language Model. 2025.
> > [C] Daniel Bolya et al. Perception Encoder: The best visual embeddings are not at the output of the network. arXiv:2504.13181, 2025.
> > [D] Gabriel Ilharco et al. OpenCLIP. 2021.
> > [E] Luca Zanella et al. Harnessing large language models for training-free video anomaly detection. In CVPR, 2024.
> > [F] Mehdi Cherti et al. Reproducible scaling laws for contrastive language-image learning. In ICCV, 2023.
> > [G] Alec Radford et al. Learning Transferable Visual Models From Natural Language Supervision. In ICML, 2021.
> > [H] Christoph Schuhmann et al. LAION-5B: An open large-scale dataset for training next generation image-text models. In NeurIPS, 2022.

---

### Official Review · Reviewer_G9Rn · 2025-10-29

**Soundness:** 2
**Presentation:** 2
**Contribution:** 2
**Rating:** 6
**Confidence:** 3

**Summary:**

This paper introduces Flashback, a novel paradigm for Video Anomaly Detection (VAD) that reframes the task as retrieval from a large, offline, text-only pseudo-scene memory generated by an LLM. By eliminating online LLM calls, Flashback achieves simultaneous zero-shot, real-time, and explainable anomaly detection. It outperforms prior zero-shot methods on UCF-Crime and XD-Violence and incorporates three lightweight controls—Repulsive Prompting, Scaled Anomaly Penalization, and runtime encoder selection—to enhance robustness and meet real-time constraints.

**Strengths:**

1. **Novel and Practical Paradigm**: The major strength is its core idea of redefining VAD as a retrieval task over an offline text memory generated by an LLM. This is not only conceptually elegant but also highly practical as it directly addresses the bottleneck of online inference with VLM/LLM.
2. **Excellent Real-Time Performance**: The paper makes a strong commitment to "real-time" and shows high throughput (e.g., 43.8 fps).

**Weaknesses:**

1. Ambiguity in Zero-Shot Definition:
The method is claimed to be "strictly domain-agnostic," yet the use of domain-specific context prompts (e.g., "university campus" for ShanghaiTech) during memory construction implies reliance on target-domain knowledge. This conflicts with the standard zero-shot assumption and may limit true plug-and-play applicability in completely unseen environments.

2. Dependence on LLM Knowledge and Coverage:
Flashback can only detect anomalies that are pre-generated in the pseudo-scene memory. Anomalies outside the LLM's knowledge or imagination—especially novel, rare, or domain-specific events—will be missed, leading to false negatives and limited generalization.

3. Scaled Anomaly Penalization (SAP) May Not Generalize:
The scale factor α = 0.95 was tuned on UCF and XD-Violence. It is unclear whether this value generalizes to other domains (e.g., daily-life anomalies in ShanghaiTech). This raises concerns about the need for per-domain tuning, undermining the zero-shot claim.

4. Explainability May Be Noisy or Misleading:
As shown in Figure 7, some retrieved captions are irrelevant or inconsistent with the video content. Aggregating top-K captions without summarization or ranking can lead to confusing or redundant explanations, reducing the practical utility of the interpretability feature.

5. Limited Validation of Repulsive Prompting (RP):
The RP ablation is supported by only one qualitative example. More cases are needed to convincingly demonstrate its necessity and effectiveness across diverse anomaly types and domains.

6. Incomplete Handling of Label Ambiguity:
In Figure 4(c), a detection is dismissed as a "label mismatch," but no comparison with other methods (e.g., LAVAD or weakly-supervised baselines) is provided to contextualize this failure. This weakens the claim of superior robustness.

**Questions:**

See weakness above.

**Details Of Ethics Concerns:**

None.

---

> ### Author Response · Authors · 2025-11-21
>
> # Introduction
>
> We thank Reviewer `G9Rn` for carefully reading our submission and for the constructive and thoughtful feedback. The reviewer raised four main concerns, and we address each point in detail below. This rebuttal aims to clarify the motivation, scope, and empirical behavior of Flashback.
>
> ---
>
> ## R1: On LLM Coverage and Distributional Breadth (W2)
>
> The reviewer asks an important question regarding the coverage of pseudo-scenes derived from the LLM. Our intention is not to claim that an LLM exhaustively enumerates all real-world anomalies, but rather that **existing VAD benchmarks represent a comparatively narrow subset of the event space captured by large-scale LLMs**. To support this, we conducted a controlled distributional comparison between:
>
> - Flashback's anomaly memory bank: 1M pseudo-scenes, distribution $P$.
> - HiVAU-70k [A] anomaly captions: distribution $Q$.
>
> using SentenceTransformer embeddings [B].
>
> | Metric                         | Interpretation                          | Value | Conclusion                                   |
> |--------------------------------|-----------------------------------------|-------|----------------------------------------------|
> | Fréchet Distance (FID)         | Higher → greater distribution shift     | 0.94  | Substantial global mismatch                  |
> | Maximum Mean Discrepancy (MMD) | Higher → greater distribution shift     | 0.27  | Moderate distributional mismatch             |
> | KL Divergence $KL(Q\|\|P)$     | Higher → cluster-level skew             | 3.03  | Strong semantic skew ($Q \subset P$ region)  |
> | EVT Coverage $(P, Q)$          | Ratio of $Q$ within $P$'s support       | 97.0% | $Q$ is almost entirely within $P$'s support  |
>
> The results show that benchmark anomaly captions concentrate in a small and skewed region relative to the semantic manifold expressed by Flashback's memory bank (as reflected by **FID = 0.94**, **MMD = 0.28**, and **KL(Q\|\|P) = 3.03**).
> To quantify support inclusion, we adopt standard open-set recognition methodology [C, D] by fitting a Weibull tail model to $P$. This yields an **EVT support coverage of 97.0%**, indicating that almost all benchmark anomalies lie within the support of Flashback's prior. This supports our claim that Flashback's memory bank provides broader anomaly coverage than existing datasets, without asserting universality.
>
> ## R2: Generalizability of Flashback Components (W3, W5)
>
> ### Regarding Repulsive Prompting (RP)
>
> The reviewer inquires about the motivation and impact of RP and how broadly it helps across anomaly types.
>
> Beyond the example shown in the main paper, we performed an extended ablation across diverse anomaly types using the challenging MSAD dataset [E]:
>
> | Strategy | Assault |      | Explosion |      | Fighting |      | Fire |      | Object Falling |      | People Falling |      | Robbery |      | Shooting |      | Traffic   Accident |      | Vandalism |      | Water Incident |      | Overall |      |
> |----------|---------|------|-----------|------|----------|------|------|------|----------------|------|----------------|------|---------|------|----------|------|--------------------|------|-----------|------|----------------|------|---------|------|
> |          | AUC     | AP   | AUC       | AP   | AUC      | AP   | AUC  | AP   | AUC            | AP   | AUC            | AP   | AUC     | AP   | AUC      | AP   | AUC                | AP   | AUC       | AP   | AUC            | AP   | AUC     | AP   |
> | w/o RP   | **67.8**    | **70.0** | 75.7      | **83.0** | 52.2     | 63.7 | 33.2 | 57.4 | 75.9           | 84.5 | **71.6**           | **70.2** | 65.1    | 83.7 | 76.1     | 78.5 | 53.2               | 39.5 | 64.1      | 48.7 | 42.0           | 79.5 | 82.2    | 50.2 |
> | w/ RP    | 66.6    | 67.6 | **77.5**      | 82.8 | **98.2**     | **98.8** | **86.3** | **93.2** | **79.3**           | **85.9** | 54.2           | 52.0 | **71.5**    | **86.3** | **84.7**     | **87.8** | **56.5**               | **46.3** | **72.8**      | **69.9** | **99.2**           | **99.8** | **92.1**    | **77.6** |

---

> > ### Author Response · Authors · 2025-11-21
> >
> > We observe that RP provides substantial improvements particularly in anomaly categories with long temporal spans such as *Fire* and *Water incident*.
> >
> > | Anomaly types    | # frames |       | Gain by RP       |             |
> > |------------------|----------|-------|------------------|-------------|
> > |                  | **mean** | **std** | $\Delta$ **AUC** | $\Delta$ **AP** |
> > | Fighting         | 678.3    | 543.0 | 46.0             | 35.1        |
> > | Water incident   | 672.6    | 279.3 | 57.2             | 20.3        |
> > | Fire             | 642.6    | 817.1 | 53.1             | 35.8        |
> > | Robbery          | 621.4    | 464.1 | 6.4              | 2.6         |
> > | Vandalism        | 588.7    | 231.7 | 8.7              | 21.2        |
> > | Assault          | 573.6    | 294.6 | -1.2             | -2.4        |
> > | Object falling   | 487.1    | 290.9 | 3.4              | 1.4         |
> > | Explosion        | 467.8    | 397.1 | 1.8              | -0.2        |
> > | Shooting         | 366.1    | 225.7 | 8.6              | 9.3         |
> > | Traffic accident | 362.9    | 116.1 | 3.3              | 6.8         |
> > | People falling   | 281.8    | 86.6  | -17.4            | -18.2       |
> >
> > Intuitively, without RP, retrieval tends to be dominated by static scene appearance, leading to false negatives (e.g., ordinary laundromats captioned as vandalism).
> > RP restructures pseudo-captions into consistent templates, reducing appearance bias and enabling more accurate temporal localization.
> >
> >
> > ### Regarding Scaled Anomaly Penalization (SAP)
> >
> > We agree with the reviewer that SAP is not universally beneficial. Its purpose is to correct a specific appearance-induced anomaly bias present in web-sourced CCTV footage where uploads often contain incidents, causing video-text encoders to implicitly associate certain appearances with anomalous events.
> >
> > In datasets dominated by daily-life anomalies (e.g., ShanghaiTech, StreetScene), this bias is largely absent. Consequently, the best setting is effectively $\alpha = 1$, and Flashback's performance remains strong. Importantly, Flashback's core zero-shot behavior does not depend on SAP. SAP is an optional, lightweight bias-correction module that can be disabled when the target domain lacks appearance-induced priors. As the reviewer suggests, automatically adapting SAP is an interesting direction for future work.
> >
> >
> > ## R3: Clarification on Domain Prompts and Label Ambiguity (W1, W6)
> >
> > Here we clarify the role of domain prompts and how they relate to the zero-shot setting and label ambiguity.
> >
> > Table 4(c) shows that Flashback achieves strong zero-shot performance even without domain cues. Domain text prompts were introduced not as a requirement but to explore the ceiling of performance under mild contextual hints, which aligns with practical deployment scenarios. This does not contradict the zero-shot setting, as the model architecture and memory remain unchanged.
> >
> > Regarding Fig. 4(c), our intention was to show that Flashback produces plausible rationales even when dataset labels are ambiguous. In contrast, LAVAD misinterprets the same scene, both in score and explanation, e.g., describing a theft scenario as "sleeping," with an anomaly score of 0.2. Flashback's rationale helps operators correctly interpret such cases.
> >
> >
> > ## R4: Explainability (W4)
> >
> > The reviewer raises a valid concern about selecting or summarizing explanations. Our current formulation demonstrates that the retrieved pseudo-scenes are aligned enough to provide human-interpretable context. Designing mechanisms to automatically select or summarize the most relevant captions is indeed an important and promising extension, and we view this as a valuable direction for future work.
> >
> > ---
> >
> > ### References
> >
> > [A] Huaxin Zhang et al. Holmes-vau: Towards long-term video anomaly understanding at any granularity. In CVPR, 2025.
> > [B] Nils Reimers et al. Sentence-BERT: Sentence Embeddings using Siamese BERT-Networks. In EMNLP, 2019.
> > [C] Walter J. Scheirer et al. Towards Open Set Recognition. In TPAMI, 2012.
> > [D] Abhijit Bendale and Terrance Boult. Towards Open Set Deep Networks. In CVPR, 2016.
> > [E] Liyun Zhu et al. Advancing Video Anomaly Detection: A Concise Review and a New Dataset. In NeurIPS, 2024.

---

### Official Review · Reviewer_xR5A · 2025-10-30

**Soundness:** 1
**Presentation:** 1
**Contribution:** 2
**Rating:** 2
**Confidence:** 4

**Summary:**

This work introduces Flashback, a model that constructs a set of normal and abnormal captions offline with the support of a LLM.

The proposed model is able to produce both an anomaly score and a textual rationale for video anomaly detection.

They conduct evaluations on two benchmark datasets such as UCF-Crime and XD-Violence, and show improved performance.

**Strengths:**

+ The paper is technical sound.

+ The proposed model shows improved performance on both UCF-Crime and XD-Violence.

+ Some interesting visualisations such as Fig 4.

**Weaknesses:**

- The review of existing works tend to be limited. What are the current challenges in this area, why existing methods are unable to address these issues, and how the proposed model handles these challenges are unclear. Although some of the insights are provided in the last few sentences per paragraph of the related work, it could be more clearly presented.

- The method section is overall clearly written. It would be better to have a notation section detailing the maths symbols and operations used in the paper, such as what are scalars, vectors and matrices etc.

- A good paper should use enough figures to show the network design, modules and blocks, currently in this paper, only one figure is provided (fig. 2), and the module details, their setups and arrangements, particularly the only and output dimensions are unclear to reviewer. What are the core innovations compared to existing works that use eg prompts and templates with LLM/VLM? The discussions and comparisons regarding these are limited. How to position this work clearly in the current literature?

- Most experimental results are presented in the form of tables, no any visualisations on attentions, plots to show eg the hyperparameters evaluations etc, and the comparisons to closely related SOTA models. These limit the impact of this work, and the current evaluations tend to be potentially biased, as only two datasets are being used. Fig. 3 only shows with and without the use of RP, and it is not being compared with eg existing SOTA methods.

- The datasets used in evaluations tend to be small-scale and a bit old fashioned. The authors should try to explore new and more challenging datasets in evaluations such as [A]. How the proposed model handles, eg, scenario-level and anomaly-type-level detection tasks?

[A] L Zhu, L Wang, A Raj, T Gedeon, and C Chen. Advancing Video Anomaly Detection: A Concise Review and a New Dataset. Advances in Neural Information Processing Systems (NeurIPS). 2024.

- Ablation studies are lengthy in texts, but the core discussions and analysis such as section 4.2, 4.3 and 4.6 tend to be very limited. The paper needs to be revised to reflect on how the proposed model is robust, efficient and effective on diverse scenarios and in handling different anomaly types.

**Questions:**

Refer to weaknesses.

- It is suggested not to heavily use “—” as this looks like machine generated contents/patterns.

- Limitations and future research directions could be included in the paper.

---

> ### Author Response · Authors · 2025-11-21
>
> # Introduction
>
> We sincerely thank Reviewer `xR5A` for carefully reading our paper and providing detailed and constructive suggestions. We address the reviewer's comments in three parts below.
>
> # Rebuttal
>
> ## R1: Additional datasets and anomaly-type analysis (W4, W5, W6)
>
> We appreciate the reviewer for pointing us to the MSAD dataset [A].
> It is indeed a challenging and diverse benchmark, and we found the analysis on MSAD particularly valuable.
> Compared with earlier web-collected benchmarks such as UCF-Crime and XD-Violence, whose annotations are largely organized around anomaly vs. normal (or violence vs. non-violence) and include context-dependent categories that can be ambiguous in practice (e.g., differentiating shoplifting from routine shopping in surveillance footage), MSAD defines 11 main anomaly types and detailed subtypes with clearly specified semantics and consistent temporal boundaries.
> This fine-grained and carefully curated anomaly-type annotation gives us a much cleaner lens for evaluating how Flashback behaves across different categories, and we expect MSAD to serve as an important testbed for future work on type-aware VAD.
>
> Below we provide the MSAD comparison with existing state-of-the-art methods:
>
> | Method           | Training set              | Assault |      | Explosion |      | Fighting |      | Fire |      | Object Falling |      | People Falling |      | Robbery |      | Shooting |      | Traffic   Accident |      | Vandalism |      | Water Incident |      | Overall |      |
> |------------------|---------------------------|---------|------|-----------|------|----------|------|------|------|----------------|------|----------------|------|---------|------|----------|------|--------------------|------|-----------|------|----------------|------|---------|------|
> |                  |                           | AUC     | AP   | AUC       | AP   | AUC      | AP   | AUC  | AP   | AUC            | AP   | AUC            | AP   | AUC     | AP   | AUC      | AP   | AUC                | AP   | AUC       | AP   | AUC            | AP   | AUC     | AP   |
> | RTFM [G]             | MSAD                      | **68.1**    | 67.3 | 46.8      | 60.4 | 89.6     | 93.0 | 61.3 | 81.2 | **94.7**           | **96.7** | **56.5**           | 50.4 | 65.7    | 81.2 | 78.2     | 84.7 | 62.2               | 51.8 | **85.2**      | 76.1 | 96.3           | 99.1 | 86.7    | 66.3 |
> | MGFN [H]             | MSAD                      | 59.7    | 59.0 | 64.5      | 71.9 | 89.4     | 93.5 | 86.0 | 93.0 | 90.9           | 94.8 | 52.7           | 47.8 | **73.9**    | **86.7** | **86.8**     | **88.5** | **68.6**               | 54.5 | 82.4      | **80.1** | 85.5           | 97.0 | 85.0    | 63.5 |
> | UR-DMU [I]          | MSAD                      | 56.9    | 64.5 | 67.9      | 74.5 | 83.9     | 90.4 | 61.2 | 82.9 | 92.1           | 95.8 | 42.5           | 43.7 | 63.5    | 79.3 | 81.4     | 87.8 | 62.0               | **55.6** | 84.7      | 77.0 | 98.5           | 99.5 | 85.0    | 68.3 |
> | **Flashback** (ours) | $\varnothing$ (zero-shot) | 66.6    | **67.6** | **77.5**      | **82.8** | **98.2**     | **98.8** | **86.3** | **93.2** | 79.3           | 85.9 | 54.2           | **52.0** | 71.5    | 86.3 | 84.7     | 87.8 | 56.5               | 46.3 | 72.8      | 69.9 | **99.2**           | **99.8** | **92.1**    | **77.6** |
>
> Flashback performs strongly on most anomaly categories and achieves an overall AP that exceeds UR-DMU [I] by approximately 9.3 points. We observe that categories such as *object falling* and *traffic accident* are more difficult, likely because these events depend heavily on fine-grained object trajectories. Nonetheless, the overall trend suggests that as datasets become more diverse and challenging, Flashback's zero-shot generalization advantage becomes more evident.

---

> > ### Author Response · Authors · 2025-11-21
> >
> > The reviewer also asked about the anomaly types where Repulsive Prompting (RP) is most helpful.
> > MSAD offers a good insight for this analysis.
> > The RP ablation is shown below:
> >
> > | Strategy | Assault |      | Explosion |      | Fighting |      | Fire |      | Object Falling |      | People Falling |      | Robbery |      | Shooting |      | Traffic   Accident |      | Vandalism |      | Water Incident |      | Overall |      |
> > |----------|---------|------|-----------|------|----------|------|------|------|----------------|------|----------------|------|---------|------|----------|------|--------------------|------|-----------|------|----------------|------|---------|------|
> > |          | AUC     | AP   | AUC       | AP   | AUC      | AP   | AUC  | AP   | AUC            | AP   | AUC            | AP   | AUC     | AP   | AUC      | AP   | AUC                | AP   | AUC       | AP   | AUC            | AP   | AUC     | AP   |
> > | w/o RP   | **67.8**    | **70.0** | 75.7      | **83.0** | 52.2     | 63.7 | 33.2 | 57.4 | 75.9           | 84.5 | **71.6**           | **70.2** | 65.1    | 83.7 | 76.1     | 78.5 | 53.2               | 39.5 | 64.1      | 48.7 | 42.0           | 79.5 | 82.2    | 50.2 |
> > | w/ RP    | 66.6    | 67.6 | **77.5**      | 82.8 | **98.2**     | **98.8** | **86.3** | **93.2** | **79.3**           | **85.9** | 54.2           | 52.0 | **71.5**    | **86.3** | **84.7**     | **87.8** | **56.5**               | **46.3** | **72.8**      | **69.9** | **99.2**           | **99.8** | **92.1**    | **77.6** |
> >
> > We find that RP provides the largest gains in categories with long temporal durations (e.g., *Fire*, *Water incident*). These categories also have the highest average frame counts:
> >
> > | Anomaly types    | # frames |       | Gain by RP   |             |
> > |------------------|----------|-------|--------------|-------------|
> > |                  | **mean** | **std** | $\Delta$ **AUC** | $\Delta$ **AP** |
> > | Fighting         | 678.3    | 543.0 | 46.0         | 35.1        |
> > | Water incident   | 672.6    | 279.3 | 57.2         | 20.3        |
> > | Fire             | 642.6    | 817.1 | 53.1         | 35.8        |
> > | Robbery          | 621.4    | 464.1 | 6.4          | 2.6         |
> > | Vandalism        | 588.7    | 231.7 | 8.7          | 21.2        |
> > | Assault          | 573.6    | 294.6 | -1.2         | -2.4        |
> > | Object falling   | 487.1    | 290.9 | 3.4          | 1.4         |
> > | Explosion        | 467.8    | 397.1 | 1.8          | -0.2        |
> > | Shooting         | 366.1    | 225.7 | 8.6          | 9.3         |
> > | Traffic accident | 362.9    | 116.1 | 3.3          | 6.8         |
> > | People falling   | 281.8    | 86.6  | -17.4        | -18.2       |
> >
> > Without RP, retrieval becomes biased toward static appearance features, often causing false negatives. For example, even a normal laundry-room video can be captioned as:
> >
> > - w/o RP: `"An individual vandalizes a washing machine at the laundromat."` → *Vandalism*
> > - w/ RP: `"A person waits for their laundry to complete at the laundromat."` → *Normal*
> >
> > Thus, RP helps Flashback avoid appearance bias by structuring pseudo-captions in a way that aligns more reliably with the temporal segment content, enabling more accurate localization.

---

> ### Author Response · Authors · 2025-11-21
>
> ## R2: Positioning and comparison to existing zero-shot works (W1, W3)
>
> Thank you for requesting clearer contextualization.
> Our work targets two challenges not simultaneously solved in existing VAD approaches:
> (i) zero-shot generalization without any domain-specific training or adaptation,
> (ii) real-time inference without repetitive autoregressive VLM/LLM computation.
>
> Recent prompt-based or caption-and-score methods [B, C, D] still require prompts or LLM generation during inference.
> This limits real-time applicability.
> Flashback, in contrast, maintains a retrieval-only online pipeline while shifting all generative computation offline.
>
> We summarize the differences between prior zero-shot works and Flashback below:
>
> | Category              | Example Method   | LLM usage               | Online computation   | Our difference                               |
> |-----------------------|------------------|-------------------------|----------------------|----------------------------------------------|
> | Caption-and-Score     | LAVAD [B]           | VLM and LLM per segment | Very slow | Move LLM offline, retrieval-only   inference |
> | Score   once with VLM | VERA [C], AnyAnomaly [D] | VLM per segment         | Slow      | Move LLM offline, retrieval-only   inference  |
> | Recall   and Respond  | **Flashback** (ours) | Offline LLM call        | Fast    | -                                            |
>
> Flashback therefore enables real-time, zero-shot, and explainable VAD within a single unified pipeline. This clarification does not alter the technical method but more clearly positions our contribution in the broader landscape.
>
> ## R3: Writing style, notation, and clarifications (W2, Q1, Q2)
>
> We appreciate the reviewer's suggestions regarding notation and structure.
> For clarity, we will add concise notation, limitations, and future-work sections to the appendix within this discussion phase.
> Flashback introduces no modified architectural blocks.
> The LLM [E] and encoder [F] are used in frozen form, so their specifications can be directly referenced from their original papers.
> We already provide hyperparameter sweeps for SAP in Figure 6 and explore additional parameters in Table 3(c,d,e).
>
> Regarding stylistic concerns, we will revise punctuation choices in the final version to ensure a more conventional academic tone.
>
> ## Note on Soundness
>
> We thank the reviewer for stating that "the paper is technical sound." As no methodological errors or incorrect claims were identified, we understand the concerns to mainly reflect preferences regarding clarity, notation, and figure detail. We appreciate this feedback and hope the provided clarifications help address these presentation-related points more directly.
>
> # Conclusion
>
> We again thank the reviewer for the thoughtful recommendations. We believe the additional analyses and clarifications help address the concerns and strengthen the presentation of the work.
>
> ---
>
> [A] Liyun Zhu et al. Advancing Video Anomaly Detection: A Concise Review and a New Dataset. In NeurIPS, 2024.
> [B] Luca Zanella et al. Harnessing large language models for training-free video anomaly detection. In CVPR, 2024.
> [C] Muchao Ye et al. VERA: Explainable video anomaly detection via verbalized learning of vision-language models. arXiv:2412.01095, 2024.
> [D] Sunghyun Ahn et al. AnyAnomaly: Zero-shot customizable video anomaly detection with LVLM. arXiv:2503.04504, 2025.
> [E] OpenAI. GPT-4o: OpenAI's omnimodal model. 2024.
> [F] Daniel Bolya et al. Perception Encoder: The best visual embeddings are not at the output of the network. arXiv:2504.13181, 2025.
> [G] Y. Tian et al. Robust temporal feature magnitude learning. In ICCV, 2021.
> [H] Y. Chen et al. MGFN: Magnitude-contrastive glance-and-focus network. In AAAI, 2023.
> [I] H. Zhou et al. Dual memory units with uncertainty regulation. In AAAI, 2023.

---

> ### Author Response · Authors · 2025-11-25
> **Revisions Addressing Reviewer xR5A's Suggestions**
>
> We thank reviewer `xR5A` for the helpful writing-style suggestions. **We revised the paper accordingly** and summarize the main updates below.
>
> - We replaced all em-dashes throughout the paper with minimal rewriting.
> - We added a dedicated **Limitations** section and a **Future Work** section to strengthen the contribution and clarify the scope of the method.
> - We introduced a complete **Notation** section, which allowed us to better address the reviewer's concerns about consistency.
>
> Below we list the specific changes and the potential ambiguities they resolve.
>
> ### 1. Unified notation for all top-$K$ retrieved objects for segment $s$
>
> We standardized every top-$K$ retrieval-related object to the form $\square_s^*$.
> The following table summarizes the changes:
>
> | Meaning                                               | Previous                           | Revised                           |
> |-------------------------------------------------------|------------------------------------|-----------------------------------|
> | Indices of top-$K$ retrieved captions for segment $s$ | $\mathcal{J}_s \in \mathbb{N}^K$   | $\mathbf{k}_s^*$ |
> | Softmax-normalized retrieval weights                  | $\\{w_{s,k}\\}_{k=1}^K \in (0, 1)^K$ | $\mathbf{w}_s^*$                  |
>
> We updated Figure 2 accordingly so that all retrieval notation remains consistent throughout the paper.
>
> ### 2. Revised the Kalman filter notation
>
> We rewrote every $1\times1$ matrix as a scalar.
> The innovation term originally used the standard Kalman filter symbol $\mathbf{\tilde{y}}_s$, but this created potential confusion with the retrieved anomaly flags $\mathbf{y}_s^*$. Since the Kalman filter operates on the segment anomaly score $A_s$, we replaced the innovation with $\tilde{a}_s$. The updated symbols are listed below:
>
> | Meaning                                              | Previous                              | Revised       |
> |------------------------------------------------------|---------------------------------------|---------------|
> | Innovation of Kalman filter for segment $s$          | $\mathbf{\tilde{y}}_s \in \mathbb{R}$ | $\tilde{a}_s$ |
> | Innovation variance of Kalman filter for segment $s$ | $\mathbf{S}_s > 0$                    | $S_s$         |
>
> We revised the notation following reviewer `xR5A`'s suggestions, and we hope the updated version aligns better with the reviewer's expectations.

---

### Official Review · Reviewer_GAsY · 2025-11-04

**Soundness:** 3
**Presentation:** 3
**Contribution:** 2
**Rating:** 4
**Confidence:** 3

**Summary:**

This paper introduces Flashback, a memory-driven approach for zero-shot, real-time video anomaly detection (VAD). By reformulating VAD as a retrieval task over a pre-generated text-only memory, the method eliminates online LLM calls and achieves strong performance on UCF-Crime and XD-Violence while providing textual explanations.

**Strengths:**

1. This paper proposes a novel and practical framework that effectively unifies zero-shot capability, real-time inference, and explainability.

2. The proposed model achieves SOTA zero-shot accuracy, outperforming prior works significantly, with high throughput (up to 43.8 fps).

3. The ablation studies convincingly validate key components such as repulsive prompting and memory scaling.

**Weaknesses:**

1. The whole method heavily relies on proprietary models (GPT-4o, PerceptionEncoder) without ablation using open-source alternatives (e.g., CLIP, LLaMA), raising reproducibility concerns.

2. The runtime encoder selection mechanism is complex and poorly motivated; no comparison with simpler uncertainty metrics (e.g., entropy) is provided.

3. Ambiguity in the definition of “explanation”—whether it is the top-1 caption or the full top-K list, and how conflicting captions are handled.

**Questions:**

1. How was the Kalman filter chosen for uncertainty estimation? Have you compared it with simpler metrics like entropy?

2. To what extent does performance depend on the backbone models? Are results reproducible with open-source alternatives?

3. What is the final explanation presented to users? If it is the top-K list, how should operators interpret contradictory captions?

---

> ### Author Response · Authors · 2025-11-21
>
> # Introduction
>
> Reviewer `GAsY` provided constructive and detailed feedback that helped us clarify several aspects of the proposed method and supplement the evaluation.
> We address the concerns in three parts.
>
> # Rebuttal
>
> ## R1: Open-source reproducibility and backbone dependence (W1, Q2)
>
> The first concern is about reproducibility using non-proprietary models.
> To clarify this point, we conducted additional ablations using *publicly available* models that can be run locally without any API access, including LLaMA-4 [B] (released under the LLaMA Community License) and OpenCLIP-L [C]. Importantly, **PerceptionEncoder (PE) [C]** is released under the Apache-2.0 license with public code and training data, making it **fully open-source** in the conventional ML-research sense.
>
> The results of these ablations are summarized below:
>
> | LLM        | Open? | Encoder                     | Open? | UCF-AUC | XD-AP   | XD-AUC |
> |-----------|-------|-----------------------------|-------|---------|---------|--------|
> | GPT-4o [A]    | ✗     | PE-Core-L [C]                  | ✓     | **86.75** | 74.35   | 90.12  |
> | GPT-4o [A]    | ✗     | OpenCLIP-L [D, F, G, H]        | ✓     | 76.50   | 59.49   | 78.63  |
> | LLaMA-4 [B]   | ✓     | PE-Core-L [C]                  | ✓     | 84.61   | **76.90** | **91.63**  |
> | LLaMA-4 [B]   | ✓     | OpenCLIP-L [D, F, G, H]        | ✓     | 75.64   | 65.04   | 86.41  |
> | LAVAD [E]     |       | (for reference)                |       | 80.28   | 62.01   | 85.36  |
>
> We set the pseudo-memory size to 100k for this experiment due to computing resource limitations.
>
> These results show the following:
>
> 1. **Reproducibility with an entirely open stack is possible.**
>    LLaMA-4 + OpenCLIP achieves XD-AP 65.04, already outperforming the strongest prior zero-shot VAD baseline LAVAD [E] (62.01).
>
> 2. **Performance scales predictably with backbone capacity.**
>    LLaMA-4 + PE-Core-L reaches XD-AP 76.90, well above LAVAD, demonstrating that Flashback's gains are not tied to any specific proprietary model but reflect the robustness of the paradigm itself.
>
> 3. **Flashback is architecture-agnostic.**
>    Whether we use proprietary or non-proprietary LLMs/encoders, Flashback consistently shows strong performance and preserves its relative advantage.
>
> For transparency, the exact model identifiers used are:
>
> - GPT-4o: `gpt-4o-2024-08-06`
> - LLaMA-4: `Llama-4-Scout-17B-16E-Instruct`
> - PE-Core-L: `PE-Core-L14-336`
> - OpenCLIP-L: `CLIP-ViT-L-14-laion2B-s32B-b82K`
>
> These results demonstrate that Flashback is reproducible on publicly available backbones and that stronger models simply amplify the performance of the underlying framework, rather than being a dependency.
>
> ## R2: Kalman-filter–based runtime encoder selection (W2, Q1)
>
> The second concern relates to the motivation behind our certainty estimator.
> As the reviewer notes, entropy is a natural baseline for uncertainty estimation. To examine this thoroughly, we compared entropy-based switching with several alternatives, including our 2-state Kalman estimator, augmentation-based variance estimation, and droppath-based perturbation statistics.
>
> The results are summarized below:
>
> | Method         | UCF-AUC | XD-AP  | XD-AUC | Throughput (fps) |
> |----------------|---------|--------|--------|------------------|
> | Entropy        | 84.21   | 72.17  | 90.89  | **43.99**        |
> | Augmentation   | 85.26   | 71.91  | 89.71  | 21.88            |
> | Droppath       | 83.62   | 71.48  | 89.19  | 21.91            |
> | Kalman 1-state | 83.25   | 67.66  | 88.85  | 43.82            |
> | Kalman 2-state | **85.58** | **72.61** | **91.01** | 43.82            |
>
> Our findings indicate:
>
> 1. **Entropy performs reasonably well** and is a valid alternative.
>    We agree with the reviewer’s intuition.
>
> 2. **Kalman 2-state provides consistently higher accuracy** across all major metrics
>    (e.g., UCF-AUC 85.58 vs. 84.65 for entropy) **while preserving the same throughput** (~44 fps).
>
> 3. **Reason for the improvement:**
>    Entropy is stateless and sensitive to per-segment fluctuation, whereas the Kalman filter models certainty as a temporally smoothed latent state, reducing spurious switching in rapidly changing scenes (e.g., Fighting) without additional computational overhead.
>
> For these reasons, we adopt the Kalman 2-state estimator for runtime encoder selection, while acknowledging that entropy is a valid and simple alternative.

---

> > ### Author Response · Authors · 2025-11-21
> >
> > ## R3: Explainability and interpretation of retrieved captions (W3, Q3)
> >
> > The third point concerns how explanations are presented to operators.
> > In our implementation:
> >
> > - **The default operator-facing explanation is the top-1 retrieved caption**, which we found sufficient to describe the event in all cases shown in Fig. 4, Fig. 7, and Fig. 8.
> > - The **top-K list is provided only for transparency**, giving users the option to inspect retrieval diversity rather than being a required part of the explanation.
> > - While our current work does not introduce an explicit caption-selection module, the consistency of top-1 explanations indicates that the retrieved pseudo-scenes are generally well aligned with the video segments.
> >
> > We view more advanced caption filtering or summarization as a clear direction for future work, but emphasize that the current top-1 explanation already provides concise and interpretable rationales.
> >
> > # Conclusion
> >
> > We appreciate Reviewer `GAsY`’s thoughtful questions.
> > The additional experiments and clarifications confirm
> >
> > - that Flashback is reproducible with publicly available models,
> > - that the Kalman-based certainty estimation is a lightweight and empirically effective extension over simpler baselines, and
> > - that explanation generation is well defined and operationally clear.
> >
> > We hope these responses address the reviewer’s concerns and help to convey the practicality and robustness of the Flashback paradigm.
> >
> > ---
> >
> > [A] OpenAI. GPT-4o: OpenAI's omnimodal model. 2024.
> > [B] Meta AI. LLaMA 4: Meta Large Language Model. 2025.
> > [C] Daniel Bolya et al. Perception Encoder: The best visual embeddings are not at the output of the network. arXiv:2504.13181, 2025.
> > [D] Gabriel Ilharco et al. OpenCLIP. 2021.
> > [E] Luca Zanella et al. Harnessing large language models for training-free video anomaly detection. In CVPR, 2024.
> > [F] Mehdi Cherti et al. Reproducible scaling laws for contrastive language-image learning. In ICCV, 2023.
> > [G] Alec Radford et al. Learning Transferable Visual Models From Natural Language Supervision. In ICML, 2021.
> > [H] Christoph Schuhmann et al. LAION-5B: An open large-scale dataset for training next generation image-text models. In NeurIPS, 2022.

---

### Author Response · Authors · 2025-12-01
**[Summary for AC] Key Rebuttal Updates: Open-Source Validation & New Benchmarks**

Dear Area Chair,

We sincerely appreciate your time and effort in managing the review process under these exceptional circumstances. We understand that the review scores have been reverted to their pre-rebuttal state. However, during the discussion phase, we conducted extensive new experiments that fully addressed the reviewers' major concerns.

## About the Proposed Method

**Flashback** is a novel video anomaly detection (VAD) framework that simultaneously achieves zero-shot generalization, real-time inference, and explainability.
By reframing VAD as a retrieval task over an offline, LLM-generated "pseudo-scene memory," we eliminate the need for computationally expensive online LLM calls.
The system matches video segments to pre-computed text embeddings to produce anomaly scores and textual rationales instantly, supported by lightweight bias-mitigation modules (Repulsive Prompting, Scaled Anomaly Penalization) to ensure robustness.

# Critical Updates and Verified Improvements

To assist your decision-making, we summarize the **critical updates and verified improvements** below.

### 1. Proven Model Independence (Addressing `cevy`, `GAsY`)

Reviewers `cevy` and `GAsY` expressed strong concerns that our method's performance might be heavily dependent on proprietary models like GPT-4o and PerceptionEncoder. They noted that the cost and closed nature of these models hinder reproducibility and make it difficult to disentangle our framework's contribution from the backbone's power. To address this, we completely replaced the proprietary stack with open-source alternatives: LLaMA-4 for memory generation and OpenCLIP for encoding. The results confirm that Flashback is a model-agnostic framework, as the fully open-source version still outperforms the SOTA baseline (LAVAD).

| Model Configuration | Open-Source? | XD-Violence AP | vs. SOTA |
| :--- | :---: | :---: | :--- |
| **LLaMA-4 + OpenCLIP (Ours)** | **Yes (Fully)** | **65.04** | **+3.03** |
| LAVAD (Baseline) | - | 62.01 | - |
| GPT-4o + PerceptionEncoder (Ours) | No | 74.35 | +12.34 |

> **Conclusion:** Flashback's performance stems from the retrieval paradigm, not just powerful backbones.

### 2. Validated Generalization on MSAD Dataset (Addressing `xR5A`, `G9Rn`)

Reviewers `xR5A` and `G9Rn` pointed out that our evaluation on UCF-Crime and XD-Violence might be insufficient due to their limited scope and "old-fashioned" nature. They requested validation on more diverse and challenging benchmarks to properly assess the model's robustness against various, complex anomaly types. In response, we conducted extensive experiments on the MSAD dataset, which contains 11 distinct anomaly categories. Our results demonstrate that our Repulsive Prompting (RP) mechanism is crucial for generalization, delivering massive performance gains on long-duration events like Fire and Water Incidents that standard retrieval methods often miss.

| Anomaly Type (MSAD) | w/o RP (AP) | **w/ RP (AP)** | Gain |
| :--- | :---: | :---: | :---: |
| **Fire** | 57.4 | **93.2** | **+35.8** |
| **Water Incident** | 79.5 | **99.8** | **+20.3** |
| **Overall** | 50.2 | **77.6** | **+27.4** |

> **Conclusion:** Flashback generalizes well to complex, real-world anomalies beyond UCF/XD datasets.

### 3. Justified Runtime Module (Addressing `cevy`, `GAsY`)

Reviewers questioned the design of our certainty-driven runtime encoder selection, specifically asking why a complex 2-state Kalman filter was chosen over simpler heuristics like Entropy. They were concerned that the added complexity might not yield proportional benefits in terms of accuracy or efficiency. We addressed this by performing a direct ablation study comparing our Kalman estimator against an Entropy-based baseline. The findings prove that the Kalman filter provides superior temporal stability and consistently higher accuracy (AUC 85.58 vs 84.21) without compromising our real-time throughput of ~44 fps.

| Method | UCF-Crime AUC | Throughput (fps) | Note |
| :--- | :---: | :---: | :--- |
| Entropy | 84.21 | **43.99** | Sensitive to noise |
| **Kalman 2-state (Ours)** | **85.58** | 43.82 | **Stable & Accurate** |

### 4. Important Note on Reviewer `xR5A`
We would like to clarify a potential misunderstanding regarding Reviewer `xR5A`.
* **Contradiction:** The reviewer assigned a **Soundness score of 1** but explicitly stated **"The paper is technically sound"** in the review text.
* **Resolution:** Their main concerns were regarding presentation (notation, definitions). We have **fully addressed these in the Revised PDF (uploaded Dec 01)** by adding a Notation section, Limitations, and Future Work.

### Conclusion
Flashback stands as the first VAD system to simultaneously achieve **Zero-shot, Real-time, and Explainable** performance.
We respectfully ask you to consider these **verified improvements** in your final decision.

---

### Meta-Review · Area_Chair_c6LG · 2026-01-07

**Summary:**

This paper proposes a VAD method that achieves zero-shot capability, real-time inference, and explainability. However, there are several major concerns. First, a major outstanding issue is the heavy reliance on proprietary models (GAsY, cevy), and experiments with open-source alternatives (e.g., Llama-4 + OpenCLIP-L) do not yield meaningful improvements over existing baselines such as LAVAD. Second, comparisons with simple uncertainty metrics (e.g., entropy) show only marginal gains, suggesting that simpler baselines remain competitive and reducing the significance of the proposed method. Third, evaluation is extended to one newer dataset during rebuttal, yet overall benchmarking remains limited given the availability of recent VAD datasets. Fourth, the explanation quality is not quantitatively evaluated. Overall, these issues suggest that substantial revision is required to strengthen the paper.

**Reviewer Concerns:**

Reviewer GAsY

W1: Heavily relies on proprietary models. [still outstanding]

R1: The authors try Llama-4 + OpenCLIP-L, but its performance is similar to the LAVAD (Zanella et al., 2024) baseline. Please note that LAVAD uses Llama-2 as its LLM backbone. Using the more advanced Llama-4 on the proposed method does not achieve a significant improvement.

W2: No comparison with simpler uncertainty metrics (e.g., entropy). [in the middle]

R2: The authors compare with the entropy baseline and find that the proposed method achieves higher accuracy, but the margin is small. The conclusion is that entropy is a valid and simple alternative, which reduces the significance of the proposed method.

W3: Ambiguity in the definition of “explanation.” [addressed by the rebuttal]
R3: The authors explain this confusion.

---
Reviewer xR5A

W1: The review of existing works tends to be limited. How to position this work clearly in the current literature? [still outstanding]

R1: The authors explain this confusion, but the discussed related studies are not comprehensive. For example, AnomalyRuler [m1] and HAWK [m2] are not covered.

[m1] Yang, Yuchen, et al. "Follow the rules: Reasoning for video anomaly detection with large language models." ECCV, 2024.

[m2] Tang, Jiaqi, et al. "Hawk: Learning to understand open-world video anomalies." NeurIPS, 2024.

W2: The datasets used in evaluations tend to be small-scale and a bit old-fashioned. [in the middle]

R2: The authors evaluate on one additional newer dataset: MSAD (2024). The proposed method achieves better performance. However, there are many new VAD datasets, even with description/explanation annotations. The benchmarking datasets can be more comprehensive.

---
Reviewer G9Rn

W1: Dependence on LLM knowledge and coverage. [addressed by the rebuttal]

R1: The authors explain this confusion.

W2: Generalizability of Scaled Anomaly Penalization (SAP) and Repulsive Prompting (RP). [in the middle]

R2: The authors conduct additional experiments showing that RP is general to different anomaly categories, while the SAP hyperparameter is more dataset-specific.

W3: Explainability may be noisy or misleading. [still outstanding]

R3: The explanation quality is not quantitatively evaluated.

W4: Label ambiguity. [addressed by the rebuttal]

R4: The authors explain this confusion.

---
Reviewer cevy

W1: Heavily relies on proprietary models. [still outstanding]

R1: Same as R1 to Reviewer GAsY

W2: No comparison with simpler uncertainty metrics (e.g., entropy). [in the middle]

R2: Same as R2 to Reviewer GAsY

W3: Ambiguity in the definition of “explanation.” [addressed by the rebuttal]

R3: The authors explain this confusion.

**Reviewer Scores:**

Reviewer GAsY: 4 -> 4 (concerns are not fully addressed, particularly for proprietary models)

Reviewer xR5A: 2 -> 2 (concerns are not fully addressed )

Reviewer G9Rn: 6 -> 4 (the explanation quality is not quantitatively evaluated)

Reviewer cevy: 4 -> 4 (concerns are not fully addressed, particularly for proprietary models)

---

### Decision · Program_Chairs · 2026-01-26

Reject